# Dissecting the effect of single- and co-infection of TB and COVID-19 pathogens on the sputum microbiome

Brintha Vijayakumar Padmavathy,[1,2,3] Ashok Kumar Shanmugavel,[4] Sivakumar Shanmugam,[4] Manikandan Narayanan[1,2,3]

**ABSTRACT** Tuberculosis (TB) and COVID-19 are both respiratory diseases, and understanding their interaction is important for effective co-infection management. Although some studies have investigated TB and COVID-19 co-infection in terms of immune responses, microbial dysbiosis in such cases remains unexplored. In this study, we understand the interface between TB and COVID-19 by systematically inspecting the microbial composition of sputum samples collected from four groups of individuals: TB only, COVID-19 only, and both TB and COVID-19 (TBCOVID) infected patients, and uninfected group (Controls). Besides metagenomic analysis of the microbiome of these sputum samples, we also performed whole-genome sequencing analysis of a subset of TB-positive samples. Different bioinformatic analyses ensured data quality and revealed significant differences in the microbial composition between Control vs disease groups. To understand the effect of COVID-19 on TB, we compared TBCOVID vs TB samples and observed (i) higher read counts of TB-causing bacteria in the TBCOVID group, and (ii) differential abundance of several taxa, including *Capnocytophaga gingivalis*. Functional profiling with PICRUSt2 revealed elevated pathways, including the pulmonary surfactant lipid metabolism pathway (with a fold change of 7.46) in the TBCOVID group. Further clustering of these pathways revealed a sub-cluster of individuals with adverse treatment outcomes. Two individuals in this sub-cluster had a respiratory pathogen, *Stenotrophomonas maltophilia*—knowing such information on key respiratory pathogens in a patient can help personalize the patient's antibiotic regimen. Overall, our study reveals the effect of COVID-19 on the airway microbiome of TB patients and encourages the use of co-microbial/co-pathogen profiling to personalize TB treatment.

**IMPORTANCE** The community of microbes in an individual's airway tract can play a complex role in respiratory diseases like tuberculosis (TB) and COVID-19. Although changes in microbial composition in TB and COVID-19 patients have been studied separately, we present a first-of-its-kind investigation of the airway-tract microbiome of individuals simultaneously infected with TB and COVID-19 pathogens. Our results highlight that co-infection with COVID-19 in TB patients alters the abundance of certain bacterial species and their related pathways. For instance, *Capnocytophaga gingivalis* is abundant in co-infected patients, but not in TB-only patients. This species and other differentially abundant species we identified in the co-morbid condition, if replicated in independent cohorts, can help explain how COVID-19 could exacerbate the severity of lung infection in TB patients. Our study also stimulates future longitudinal studies using expanded data sets to understand the role of concomitant pathogens and assess whether adjusting the antibiotic regimen accordingly can improve TB treatment outcomes.

**KEYWORDS** tuberculosis, COVID-19, SARS-CoV-2, lung infection, co-infection, co-morbidity, differential abundance, functional profiling, metagenomics, microbial composition, microbiome

**Peer Reviewers** Robert Matovu, Makerere University, Kampala, Kampala, Uganda; Neeta Pradhan, Johns Hopkins India Pvt Ltd and B.J Medical College Clinical Research Unit, Pune, Maharashtra, India; Bingdong Zhu, Lanzhou University, Lanzhou, Gansu, China

Address correspondence to Sivakumar Shanmugam, shanmugam.sk@icmr.gov.in, or Manikandan Narayanan, nmanik@cse.iitm.ac.in.

Sivakumar Shanmugam and Manikandan Narayanan contributed equally to this article.

The authors declare no conflict of interest.

See the funding table on p. 19.

Tuberculosis (TB) is an airborne respiratory disease affecting millions of people worldwide (1). COVID-19, another airborne respiratory illness, has caused a global pandemic and disrupted healthcare systems, including progress toward TB eradication. TB is caused by an ancient bacterial pathogen, *Mycobacterium tuberculosis* (*M. tb*), whereas COVID-19 is caused by the novel SARS-CoV-2 virus. Though the pathogenesis of these diseases is different, they could synergistically increase host inflammatory responses in the lung (2) and also disrupt the host's microbial environment in the airway tract (3, 4).

Understanding the interplay of these diseases can yield new insights into developing new treatment regimens for co-infection with TB and COVID-19 (referred to as TBCOVID hereafter), as well as tackling similar disease combinations in the future. Concurrent infections of TB and COVID-19 could worsen the patient's condition and complicate recovery (5, 6), but available clinical data on TBCOVID co-infection is too limited to understand the underlying mechanisms (7). One study found that TB hinders SARS-CoV-2-specific immune responses in co-infected individuals (8), while mouse models show that the presence of bacterial loads inhibits further viral replication (9, 10). Apart from determining the immune responses, studying the microbial composition in the human airway tract can also reveal new insights into the functional interactions between TB and COVID-19.

The human body hosts a diverse set of microbes, and numerous studies have linked species/genus in this human microbiome with different diseases, affecting both disease severity and treatment outcomes (11, 12). Most existing metagenomic studies have been conducted separately on either TB or COVID-19 cohorts, with none focusing on TBCOVID co-infections. For instance, different studies have analyzed the microbial composition of sputum, gut, and lung samples from TB patients (3, 13–15), providing insights into microbial alterations between TB and healthy samples. Similarly, the metagenomic analysis of nasopharyngeal microbiota from COVID-19 patients reveals the presence of opportunistic pathogens and depletion of oral microbiome diversity (4, 16). As the microbiota undergo severe dysbiosis in cases of co-infection, an in-depth analysis of the microbial composition between different single-infection and co-infection groups can aid in the identification of new double-disease signatures of clinical relevance (17). The new signatures could not only facilitate medical prognosis but also contribute to the development of new therapeutic interventions (18).

In this study, we performed a systematic metagenomic and whole-genome sequencing (WGS) analysis of sputum samples (see Fig. 1) to uncover the microbial signatures associated with diseases, such as TB, COVID-19, and especially their co-infection. Sputum samples were collected from four groups—TB (only), COVID-19 also referred to as COVID (only), TBCOVID, and uninfected Controls. Quality and batch effect checks were performed prior to downstream analyses. Differential abundance (DA) analysis revealed elevation of pulmonary disease-associated taxa in the TB and TBCOVID groups relative to the Control group. Moving from single- to double-disease analysis, we detected species, such as *Capnocytophaga gingivalis*, *Escherichia coli*, *Prevotella melaninogenica*, and *Veillonella parvula*, to be significantly elevated in the TBCOVID group compared to the TB only group. Further investigation of the microbial community pathways revealed dysregulation of pulmonary lipid-related pathways in the TBCOVID group, potentially driven by the presence of pathogens, such as *Stenotrophomonas maltophilia*. Clustering of individuals by the top differentially active pathways, when considered alongside their disease statuses retrieved from the clinical portal after more than 3 years post-sample collection, provided insights into the impact of COVID-19 on adverse outcomes in TB treatment. Analysis of the WGS reads of *M. tb* from a subset of TB and TBCOVID samples showed no significant differences in the genetic composition between the TB and TBCOVID groups. In summary, our comprehensive analysis not only revealed alterations of the lung microbiome (and associated species, genera, pathways, and genes) in the presence of single/co-infections, but also highlighted how to use

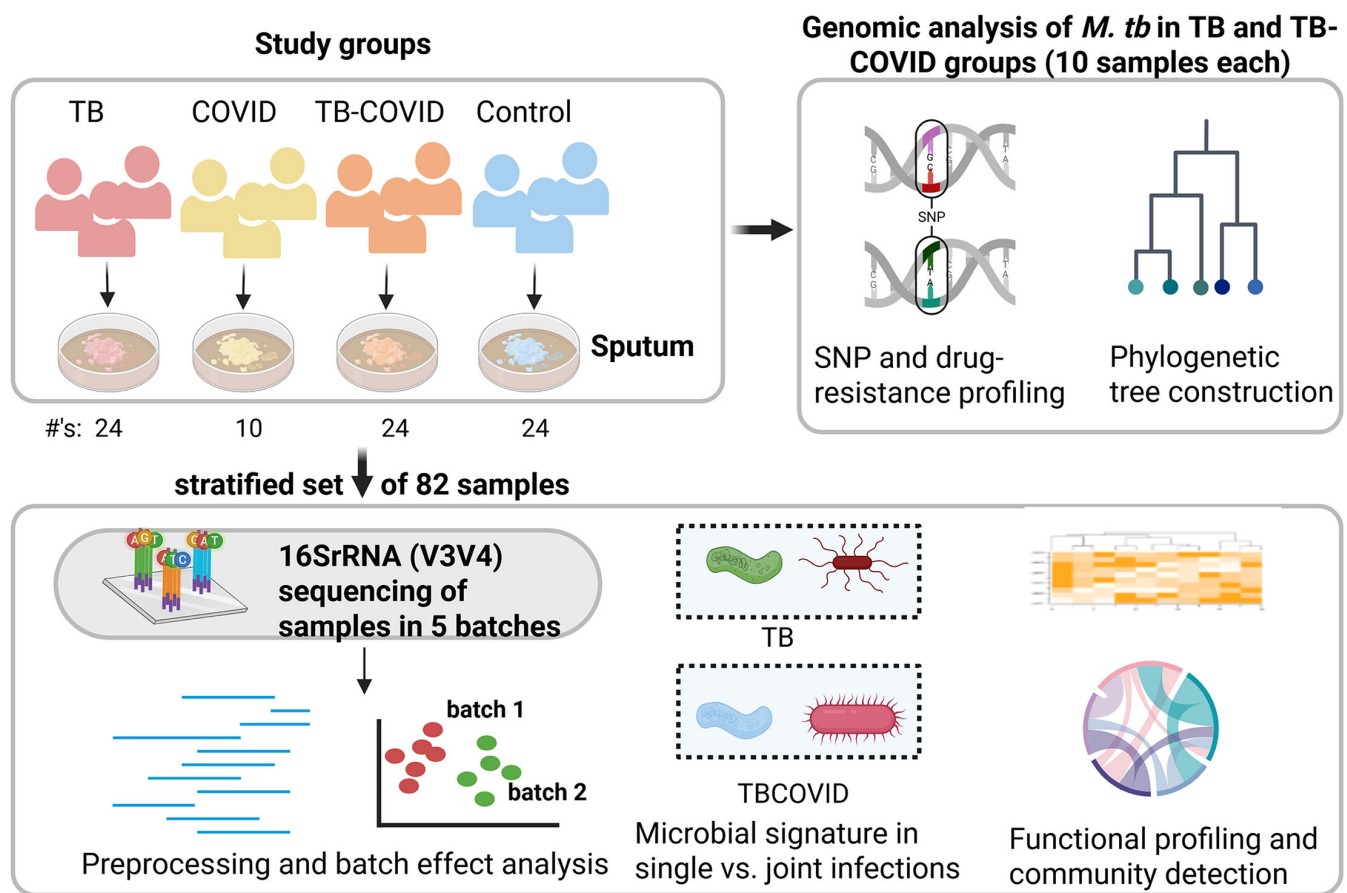

**Research Questions**

1. The sputum microbiota undergo dysbiosis in the presence of diseases like TB or COVID-19. How does this dysbiosis differ in cases of TB and COVID-19 co-infection?

2. Does the functional profile of sputum microbiota differ during TB/COVID-19 co-infection?

**Study groups**

TB    COVID    TB-COVID    Control

Sputum

#'s:  24       10         24          24

stratified set of 82 samples

**Genomic analysis of *M. tb* in TB and TB-COVID groups (10 samples each)**

SNP and drug-resistance profiling

Phylogenetic tree construction

16SrRNA (V3V4) sequencing of samples in 5 batches

batch 1

batch 2

Preprocessing and batch effect analysis

TB

TBCOVID

Microbial signature in single vs. joint infections

Functional profiling and community detection

**Metagenomic analysis**

FIG 1 Overview of our double-disease multi-omic study. In this study, we collected sputum samples from four different groups (TB, TBCOVID, COVID, and Control) and analyzed a stratified subset of them via metagenomic (specifically 16S rRNA [V3V4]) sequencing. Additionally, we subjected a subset of TB and TBCOVID samples to WGS sequencing of the TB genome. Different bioinformatic analyses were performed to ensure the quality of the generated data and to understand the impact of COVID-19 on TB patients with respect to microbial composition and TB strain characteristics.

the detected microbes/pathogens in an individual to potentially personalize treatment strategies.

## MATERIALS AND METHODS

### Sample collection, covariates' information, and selection procedure

Ethics approval was obtained for a cross-sectional study involving four groups of participants: TB, TBCOVID, COVID, and Controls. Sputum samples were collected during the first visit of presumptive TB patients to the hospital, and a follow-up was conducted only for confirmed TB patients (in the TB and TBCOVID groups), not for additional sample collection, but to assess their clinical treatment outcomes. This follow-up, with a median duration of 6 months after the first visit, was carried out by independent healthcare

providers (not the authors of this study) and recorded in the Nikshay portal (an online TB management and patient-tracking system for healthcare providers in India). We accessed the Nikshay portal during March 2022 to retrieve TB patients' covariate metadata and in February 2025 to obtain TB patients' latest outcome status prior to manuscript submission. All the participants involved in the study provided their consent for data collection.

Presumptive TB patients were tested for TB using smear grading acid-fast bacillus (AFB) test and the cartridge-based nucleic acid amplification test (CBNAAT) performed on the GeneXpert platform (19). The outcome of the CBNAAT test was used to determine whether the patient had TB or not. All the participants with/without TB were tested for SARS-CoV-2 infection at NIRT during 2021. A total of 1,132 sputum samples (461 from the group of TB patients) were collected and screened for SARS-CoV-2 infection using the STANDARD Q COVID-19 Ag Test (SD Biosensor) kit. The samples (32 out of 461) that tested positive for COVID-19 from TB patients were classified as the TBCOVID group and further filtered as described below. The samples (10) from non-TB patients that tested positive for COVID-19 were designated as the COVID group and subjected to metagenomic sequencing and further analyses. The remaining samples from non-TB patients were categorized as the Control group. To maintain balance in downstream analyses, we selected 24 samples from each of the TB, TBCOVID, and Control groups (as explained in detail below) and subjected them to metagenomic sequencing and further analyses.

The metadata of the TB patients extracted from the Nikshay portal includes different covariates, such as age, gender, socioeconomic status, new/previously treated, weight, height, treatment phase, treatment outcome, HIV status, diabetic/non-diabetic, tobacco smoking, alcohol consumption, smear, and other TB drug resistance-related ones. More information on these covariates, such as the different values each covariate takes and their corresponding summaries, can be seen in Table S1 and Supplementary File D1a. For instance, the smear covariate of a TB patient can take any of three values based on the AFB sputum smear grading test, provided the test result is positive: 1+ positive, 2+ positive, or 3+ positive. These smear grade values are based on the bacilli load (with 1+ positive indicating lower bacilli load and 3+ positive indicating higher load) and hence can be taken as a proxy of TB severity (20, 21). We would like to note that for all the TB and TBCOVID patients profiled in this study, both AFB and CBNAAT tests agreed in their outcomes—specifically, all CBNAAT positive patients were also AFB-positive (see Supplementary File D1b). These covariates and their subsets were used for subsequent analyses as follows:

- A set of 21 covariates was used with the propensity score matching (PSM) method to perform sample selection (specifically, all covariates listed above, after excluding redundant covariates pertaining to test/clinical interpretation of drug resistance and replacing height and weight covariates with a single covariate, body mass index [BMI], to simplify the PSM method's application).
- A set of nine covariates was included for covariate adjustment in statistical models used to assess differential abundance and other effects between TB vs TBCOVID groups. Specifically, these covariates were age, gender, weight, height, diabetic/non-diabetic, tobacco smoking, alcohol consumption, smear, and sequencing batch number. Note that the smear covariate capturing *M. tb* load (based on an independent sputum smear grading test) was used for covariate adjustment, so as to understand the impact of COVID-19 on the microbial composition of TB patients with similar *M. tb* loads. COVID-19 may influence the microbial composition of TB either by increasing the *M. tb* load or via other mechanisms—this study focuses on the latter by adjusting for the smear covariate. Note also that we excluded drug resistance-related covariates from covariate adjustment and other downstream analyses, since all selected individuals, except for one, did not exhibit any drug resistance.

- Just one covariate, sequencing batch number, was used to adjust the data of disease vs Controls analysis in statistical models, because the rest of the above nine covariates were unavailable for the Control group (and also for the COVID group, as these non-TB individuals were not part of the Nikshay portal).

In the TB and TBCOVID groups, first filters were applied to exclude samples from previously treated patients, and those with different treatment regimens, missing weight attributes, or unknown HIV status. Samples with missing height values were retained, resulting in 314 samples after filtering. The missing height values were imputed by using the mean of the height values from the remaining samples. A new covariate, BMI, was added by calculating weight (kg)/height$^2$ (m$^2$). Ignoring height and weight, remaining covariates (including BMI) were used to select 24 samples each for TB and TBCOVID groups using the PSM method (22). The male and female categories were inadvertently grouped into a single category when conducting PSM (nevertheless, Fig. S1 shows that the composition of differentially abundant species reported in the Results section on "Effects of COVID-19 on TB" is not biased by the gender covariate). PSM was implemented using functions of the psmpy package in Python (23). Specifically, 24 samples from the TBCOVID group were initially chosen based on height attribute (22 samples with available height values were selected, and the remaining two samples with missing height values were chosen randomly). For each member of the TBCOVID group, the closest neighbor in the TB group was identified using the KNN matching algorithm of the psmpy package based on their propensity scores. The summary statistics for the age, height, and weight covariates of matched TB and TBCOVID samples selected using the PSM method are shown in Table S1. As the covariates of non-TB groups were not collected, all 10 samples from the COVID category and a random subset of 24 samples from the Control category were selected for further analysis. Our primary focus is to identify the differences in the microbial composition and their interactions through network analysis between the TB vs TBCOVID groups, with the COVID and Control groups serving as baselines.

## DNA extraction procedure, library preparation, and sequencing

The library preparation and metagenomic sequencing of the selected samples were carried out at MedGenome, Bengaluru, in five batches (see Table S2 for details). Libraries for V3–V4 amplicon sequencing were prepared using the NEBNext Ultra DNA Library preparation kit. The amplicons were subjected to a sequence of enzymatic steps for repairing the ends and tailing with deoxyadenosine (dA), followed by ligation of adapter sequences. These adapter ligated fragments were then cleaned up using solid-phase reversible immobilization beads. The clean fragments were indexed during a limited-cycle polymerase chain reaction to generate final libraries for paired-end sequencing. The resulting libraries were quantified before loading on the cBot for cluster generation and sequenced on the Illumina MiSeq system platform. WGS was also performed at MedGenome using Twist MF Kit—104177 for library preparation and sequencing on Novaseq Xplus- 2X150 platform.

## Preprocessing, taxonomy assignment, functional and pathogen profiling

The raw reads (37,221,621 in total) were checked for quality using the FastQC tool (v0.11.9) (24). Trimmomatic (v0.39) was employed for universal adapter removal and read trimming using a sliding window of 4 with a quality threshold set to 20 (25) to retain the majority of high-quality data. The forward (CCTACGGGNGGCWGCAG) and reverse (GACTACHVGGGTATCTAATCC) primers present at the beginning of the reads were removed using the cutadapt (v4.9) tool with the maximum error rate option set to 0.2. Trimmed reads (36,560,363 reads, which is 98.2% of the total raw reads) were denoised using the DADA2 tool (v1.26.0) (26), resulting in a set of denoised sequences known as amplicon sequence variants (ASVs). Filtering of low-quality reads and chimera

removal was also performed as part of the denoising process, resulting in the retention of 50.2% of trimmed reads and the identification of 33,375 ASVs across all samples for downstream analysis (see also Supplementary File D2 for the per-sample counts of raw and filtered reads). Note that denoising was performed separately for each batch of samples, grouped by sequencing run, and recall that the sequencing run is used for covariate adjustment in statistical analyses.

For the taxonomy assignment, we chose the Human Oral Microbiome Database (HOMD) database over Greengenes and SILVA databases (27), as HOMD is specifically customized to the oral region and the related airway tract rather than the general Greengenes/SILVA databases, and HOMD can hence assign more plausible airway-tract-related taxa to ASVs. We downloaded the HOMD v10.1 from the HOMD website as Prokka-annotated Enzyme Commission numbers were available for these sequences 28). We then took only the unique sequences from this database and used the V3–V4 region of these sequences to train a naive Bayes classifier in QIIME2 (v2022.11.1) (29). The confidence cutoff for the taxonomy assignments using a naive Bayes classifier was set to 97% to precisely facilitate the identification of pathogens (30). When we applied this trained classifier to the 33,375 ASVs, 8,271 (24.8%) were assigned taxa at the genus or species level. Hence, we focused on these 8,271 ASVs alone for obtaining the genera/species read counts of each sample.

Functional profiling was performed using the PICRUSt2 tool (v2.5.2) (31) for the above set of 8,271 ASVs. The HOMD sequences used for taxonomy classification above were employed to construct the reference database for PICRUSt2, following the steps outlined by the PICRUSt2 authors. Since all but one of the input ASVs were poorly aligned to these HOMD-based reference sequences, we used the reverse complements of ASVs instead for functional profiling, as specified in reference 32 (87.8% of the 8,271 ASVs aligned to the reference sequences after applying reverse complement). The output of this functional profiling step is the estimated activity of different microbial pathways in each sample, and the ASVs contributing to these pathways.

We compiled a catalog of respiratory pathogens, primarily based on the eight common ones reported in reference 33. Of these eight, only four were present in our metagenomic data set: *Stenotrophomonas maltophilia*, *Acinetobacter baumannii*, *Klebsiella pneumoniae*, and *Haemophilus influenzae*—and were therefore included in the catalog. In addition, *Alloscardovia omnicolens* was included as the final fifth pathogen in our catalog, as it was detected in all groups except the Control group and was the only species significantly enriched in the COVID group compared to the Control group (see Results subsection "Altered genera/species in TB, TBCOVID, and COVID groups (vs Control)"). Other common respiratory pathogens like *Streptococcus pneumoniae*, *Staphylococcus aureus*, and *Pseudomonas aeruginosa* were not observed in any of the samples, and hence not included in the catalog. The distribution of pathogens in the catalog across the TB, TBCOVID, COVID, and Control groups is presented in Fig. S2.

Different downstream analyses were performed on the samples' genera/species counts and pathway activities, transformed into relative or centered log-ratio (CLR) abundances (34) (see Supplementary File D3 for the genus, species, and pathway abundances). The code for all the downstream analyses was written in R (v4.4.1), Python (v3.8.13), and Linux bash shell (available at https://github.com/BIRDSgroup/Multi-omic-TB-COVID).

## Statistical analyses

Statistical analyses were performed for the taxa identified in our metagenomic samples. Both genus-level and species-level analyses were performed. The analyzed taxa are obtained by taking the distinct set of taxa assigned to ASVs at the genus and species levels. We wrote a custom code in R (released openly at the same project GitHub link mentioned above) to perform this task—this code also prepares genus-level read counts by accumulating the read counts of ASVs belonging to a genera of interest and any species belonging to the same genera. We computed measures, such as intraclass

correlation coefficient (ICC) for Shannon's diversity (35) and the coefficient of variation (CV) (36) at the genus level to evaluate the quality of sequencing. ICC was estimated using the ICC function of the psych package (37). CV was calculated separately for the set of repeat samples of an individual (with at least three repeat samples, also known as technical replicates) to estimate the amount of technical variation, and the set of non-repeat samples from different individuals to estimate the magnitude of biological variation. Please refer to Fig. S3 for more details. Note that CV was estimated for a set of samples as per the following equation:

$$CV = \frac{\sigma}{\mu}, \tag{1}$$

where $\sigma$ is the standard deviation and $\mu$ is the mean of the relative abundance of each genus across the samples in the set. A glossary of definitions of different measures used in the statistical analysis is provided in the Supplementary section Definitions of key measures.

We used alpha and beta diversity measures to assess the taxonomic composition of samples both within and across groups. Analysis of variance (ANOVA) test was used to assess differences in the alpha diversity between groups with a false discovery rate (FDR) threshold ≤0.05. Principal coordinate analysis (PCoA) on distance matrix based on Bray-Curtis dissimilarity measure was used to visualize the between-sample diversities, and this distance matrix was also used to assess the effect of covariates on group differences using the permutational multivariate analysis of variance (PERMANOVA) test.

We performed DA analysis on the genus, species, and pathway abundances using multiple DA methods to increase the robustness of our findings. We specifically used three DA methods: analysis of compositions of microbiomes with bias correction (ANCOM-BC) (38), linear regression framework for differential abundance analysis (LinDA) (39), and correlated observations with the beta-binomial (corncob) (40). The reason for choosing ANCOM-BC and LinDA is that both methods take into account the compositional nature of the data and perform differential analysis on the bias-corrected data. Although these methods are based on bias correction, corncob is based on a beta-binomial regression model that takes into account variability in taxa counts across samples. Taxa present in more than 5% of the samples with read counts >5 were considered for the differential analysis to ensure reliable statistical results (41, 42). Taxa were reported as DA if they had an absolute log fold change (LFC) ≥1.5 as determined by LinDA (in $\log_2$ base), and also met at least one of the following criteria: (i) selected by at least two of the DA methods (ANCOM-BC, LinDA, and corncob) at an FDR threshold of 0.2 for genera/species and 0.01 for pathways; (ii) identified as structural zeros (completely absent in one group but present in the other group) by ANCOM-BC. To visualize the DA genera/species, we plot their normalized abundances in the two groups of samples being compared. Normalized abundance of a taxa in a sample is obtained by taking the natural logarithm of the taxa's read count in the sample (after adding 1), followed by subtraction of the sample-specific bias estimated by ANCOM-BC (during its comparison of the two sample groups). Note that ANCOMBC also reports LFC values in natural logarithm ($\log_e$).

The NetCoMi package (43) was used to construct and visualize the association networks using the SpiecEasi method. We also employed the linear regression framework outlined in (44) to find the genera/species, if any, that lie at the interface of both COVID-19 and TB.

## WGS data analysis

WGS data of *M. tb* was obtained from the sputum samples of 20 individuals—9 from the TB group and 11 from the TBCOVID group. We used the tool Demixer (45) developed in our lab for identifying the TB strains in each of the samples. The default version of Demixer uses a hybrid approach with the TB-Profiler (46) mutations as a reference database to identify known and novel strain identities. The following in-house pipeline

was used to generate the .vcf files for analysis with Demixer. The raw reads in each sample were mapped against the H37Rv reference genome (NCBI Reference Sequence NC000962.3) using the Burrows-Wheeler Alignment tool (v0.7.17) (47). The SAM files were sorted and then converted to BAM files using bcftools (48). Variant calling was performed using FreeBayes (v1.3.6), with Demixer's merged .vcf file as the reference (49). The resulting VCF files were merged using the bcf merge tool, and Demixer was run on the merged .vcf file to determine the strain identities (and corresponding lineages) and strain proportions in each sample. In the case of mixed infection samples that have more than one strain, the lineage of the sample is assigned based on the lineage of the majority reference strain in the sample. Here, lineage refers to a distinct evolutionary grouping of the *M. tb* bacterial strains (50). The phylogenetic tree of the consensus sequences (obtained using bcftools) was built using the NGPhylogeny workflow (51), which incorporates a set of tools, namely MAFFT, BMGE, and FastME, and was visualized using iTOL (v6) (52). We also ran TB-Profiler to get the drug resistance profile of each sample.

## RESULTS

### Clinical characteristics of the study population

For the purpose of metagenomic sequencing, a stratified subset of 82 samples was chosen systematically using the PSM method. The metadata of these samples, including the replicates, is given in Supplementary File D1b. Table S1 highlights the clinical characteristics of the two groups, TB and TBCOVID. The number of males (85%) was comparatively higher than that of females. All selected TB patients in our study were newly diagnosed cases, and none were co-infected with HIV. They received the same treatment regimen, ensuring consistency in treatment across the cohort. As the treatment outcome covariate indicates the treatment efficacy 1 year after data collection, it was excluded from further statistical analyses. Nearly 30% smoke tobacco, and 48% consume alcohol. The severity of TB was represented using the "smear" covariate, where 3+ positive indicates higher disease severity and 1+ positive indicates low disease severity. All individuals included, except for one, did not exhibit any drug resistance, and therefore, the drug-related covariates were not considered in any of the downstream analyses. Understanding the distribution of these covariates helped us select the subset of covariates to be used for covariate adjustment in statistical models of differential abundance and other effects between TB vs TBCOVID groups (see Materials and Methods).

### Quality assessment of metagenomic sequencing using replicates and mycobacterial loads

Sequencing provides relative abundances of taxa rather than absolute abundances and is subject to variability due to sampling procedures (38). Therefore, we examined the read counts of technical replicates to assess the level of variability. For instance, the scatterplot for the technical replicates of a TBCOVID sample (Fig. 2a) shows that the read counts at the genus level were significantly correlated with a correlation coefficient of at least 0.88. Hierarchical clustering based on Spearman correlation coefficients computed between the genus counts of each pair of samples (Fig. 2b) shows that the replicates of the same sample cluster closely together, except for one from the Control group. Additionally, ICC for Shannon's diversity across all replicates at the genus level was found to be 0.67, indicating moderate reliability (53) with confidence intervals ranging from 0.29 to 0.91. This suggests that the read counts across the technical replicates are reasonably reliable, and the moderate ICC may be influenced by the limited number of replicates (fewer than 3 for most samples, except two having 3 replicates). Focusing on a genus with similar mean relative abundance, the CV of the relative abundance of such a genus across biological samples was found to be higher than that across technical replicates of a sample (see Fig. S3), highlighting that biological variability dominates

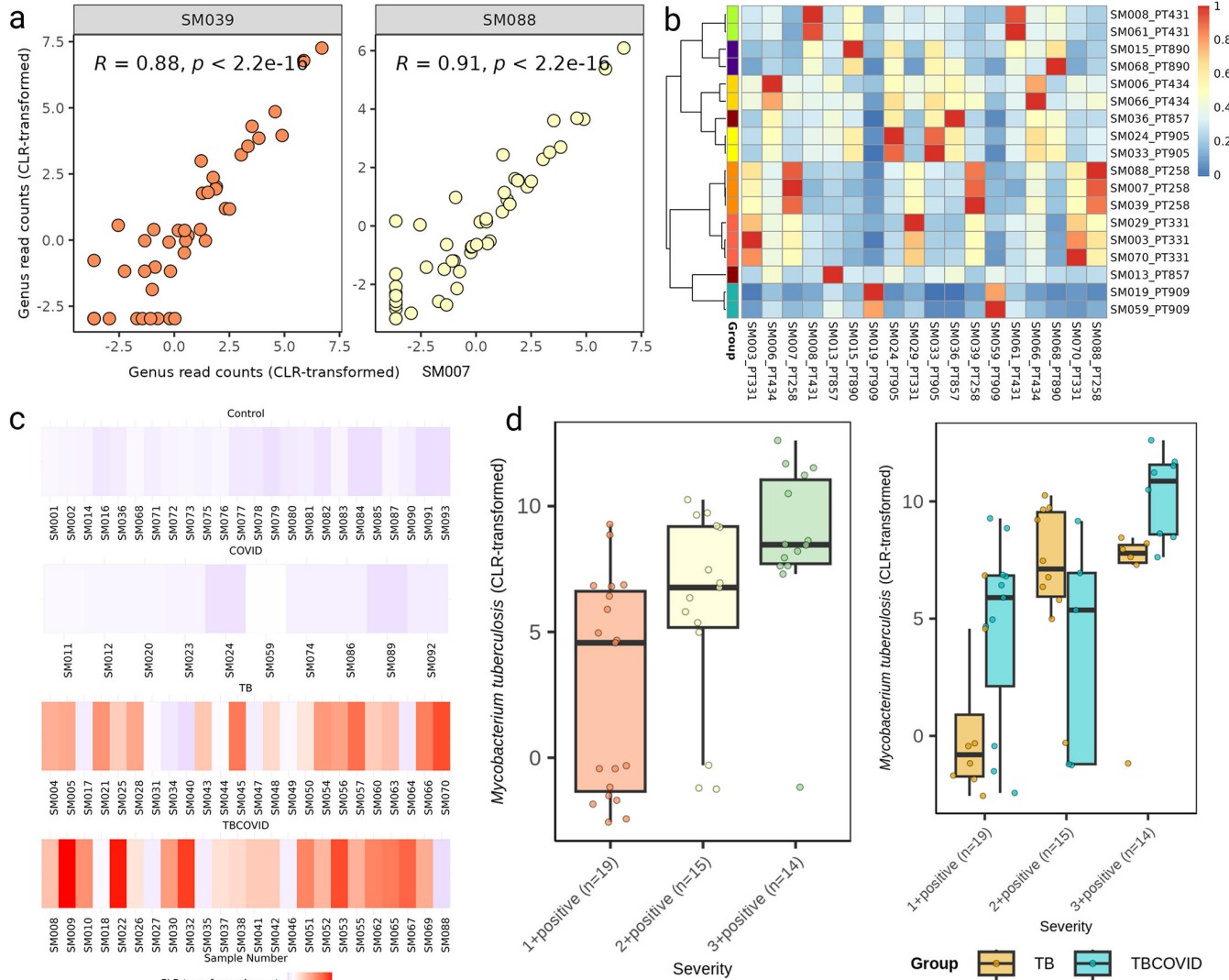

**FIG 2** Analysis of read counts for quality assessment. (a) The correlation plots of the CLR-transformed genus counts between the TBCOVID sample SM007 and its technical replicates, SM039 and SM088, with Spearman correlation coefficients (R) indicated. (b) Heatmap of the Spearman correlation coefficients computed between the genus-level read counts of each pair of replicates of samples. Rows of this heatmap are hierarchically clustered, using the pheatmap package in R with a correlation-based distance measure, and labeled so that the technical replicates from the same sample get a unique color. Columns are not clustered. (c) Heatmap of *M. tb* counts observed in each of the samples. (d) CLR-transformed read counts of *M. tb* at different levels of smear grade are shown (for the combined set of TB and TBCOVID samples in the left panel, and the separate sets in the right panel).

technical variability (as should be the case in a high-quality data set). Figure S3 also showed that taxa with smaller read counts exhibit higher variability than taxa with larger read counts, as expected (54). Based on these analyses, we selected one replicate with the highest read count at the species level for each set of technical replicates to represent the corresponding samples.

On further investigating the *M. tb* read counts obtained for each sample, we found that *M. tb* species was not observed in any of the samples in Control and COVID groups, but present in 66% of the samples in TB group and 79% of the samples in TBCOVID group (Fig. 2c). This finding further strengthens the reliability of the sequencing process and the overall quality of the samples. We also found that the smear grade-based TB disease severity has an effect on the observed (CLR-transformed) *M. tb* read counts (Fig. 2d; ANOVA *P* value 0.00047 for the boxplot shown in the left panel). Overall, the results indicate that the sequencing quality is sufficiently robust for conducting further analysis.

## Microbial abundance, diversity, and composition of the disease groups

A first look at the microbial abundances showed that *Rothia mucilaginosa* was the most abundant species based on its overall mean relative abundance in all groups, while *Escherichia coli* exhibited a higher abundance in the COVID and Control groups compared to the TB and TBCOVID groups (see Fig. 3c). At the genus level, Streptococcus was the most abundant, followed by *Rothia*, *Veillonella*, and *Haemophilus* (see Fig. S4).

We next assessed species richness across the four groups using alpha diversity measures: observed taxa, Shannon index, and Simpson index (see Fig. 3a). ANOVA results indicated no significant differences in species richness among the groups. On examining Bray-Curtis dissimilarity between the species abundances of samples from the two groups and visualizing the results with PCoA (shown in Fig. 3b), we noticed that the disease status has an effect on the clustering of samples, except in the comparison between Control and COVID groups. PERMANOVA with covariate adjustment at species level also supported these findings for comparisons between Control vs TB ($P$ value = 0.026, $R^2$ = 0.034), TB vs TBCOVID ($P$ value = 0.006, $R^2$ = 0.042), Control vs TBCOVID ($P$ value = 0.02, $R^2$ = 0.033), and finally the Control vs COVID ($P$ value = 0.665, $R^2$ = 0.024) groups. The differences in species abundances detected by PERMANOVA cannot be explained by any changes in the dispersion of these abundances (because the dispersion of different groups is similar at the species level, according to $P$ values from a dispersion analysis detailed in Supplementary Information—Control vs TB $P$ value = 0.348, TB vs TBCOVID $P$ value = 0.096, Control vs TBCOVID $P$ value = 0.415, and Control vs COVID $P$ value = 0.608). When repeating the PERMANOVA analysis using covariate-adjusted genus instead of species counts, we found no significant differences in all four comparisons— Control vs TB ($P$ value = 0.329, $R^2$ = 0.023), TB vs TBCOVID ($P$ value = 0.199, $R^2$ = 0.026), Control vs TBCOVID ($P$ value = 0.523, $R^2$ = 0.019) and Control vs COVID ($P$ value = 0.684, $R^2$ = 0.022). These findings suggest that microbial composition differs more at the species than genus level between the different groups. Note that Fig. S4 shows the genus-level alpha diversity and PCoA plots computed using abundances that were not adjusted for covariates.

To increase the robustness of the above findings, we finally wanted to test if any batch effect is present in our data set. Principal component analysis of the CLR-transformed counts data at the species/genus level (see Fig. S5) revealed no distinct clustering of

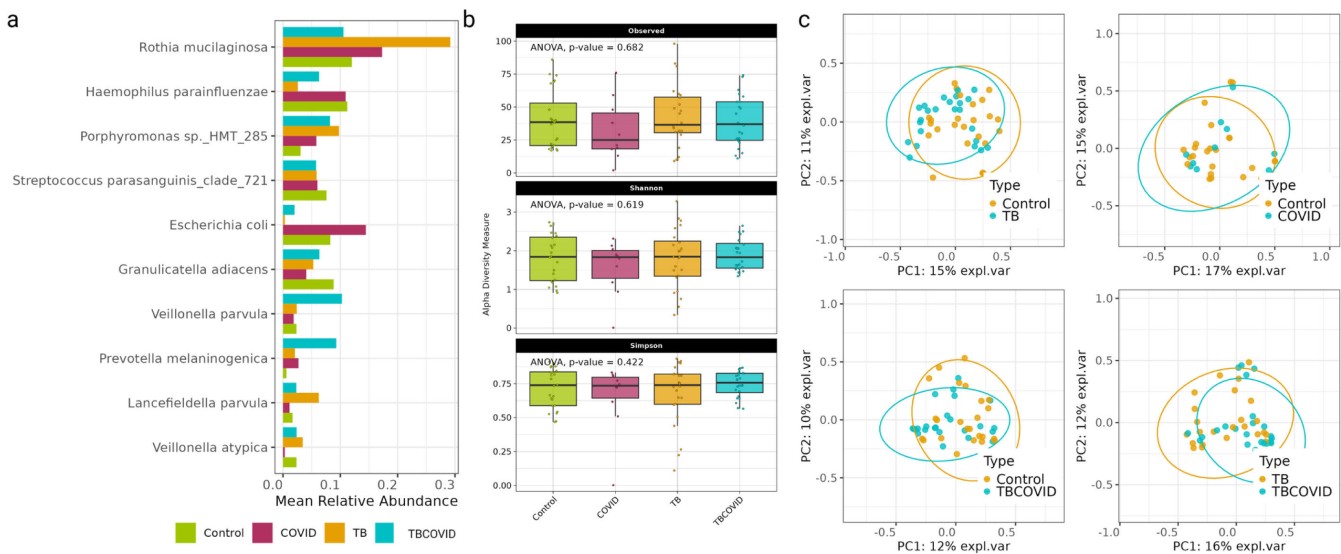

**FIG 3** Characterization of species diversity and composition in metagenomic sequencing data obtained from the four groups. (a) Visualization of the top 10 abundant species across the four groups, ranked based on the average of their per-group mean relative abundances. (b) Boxplots of the alpha diversity measures (observed taxa, Shannon index, and Simpson index) at the species level across the four groups. (c) PCoA plots based on the Bray-Curtis dissimilarity measure computed between the species abundance profiles of samples.

samples based on batch identities. Additionally, we computed the Bray-Curtis dissimilarity measure for samples within the same category for the different batches to assess compositional diversity. This analysis also did not reveal any batch effects, as diversity was largely consistent across batches (see Fig. S5).

Overall, the diversity analysis indicates that the microbial compositions differ more substantially between the groups Control vs TB, Control vs TBCOVID, and TB vs TBCOVID compared to the Control vs COVID group, at least at the species level. This also suggests that the presence of both TB and COVID together leads to distinct microbial compositions, rather than when infected with a single disease.

## Altered genera/species in TB, TBCOVID, and COVID groups (vs Control)

We then investigated how genera and species vary in the presence of diseases, such as TB, COVID-19, and both. To ensure robust biological interpretations (55) and for other related reasons explained in the Materials and Methods, we applied multiple DA tools, namely ANCOM-BC, LinDA, and corncob, on the genera/species abundance counts of Control and disease samples (analyzed one disease at a time, with sequencing run used for covariate adjustment; see Materials and Methods).

At the genus level (Fig. 4), both *Prevotella* and *Mycobacterium* were enriched in TB and TBCOVID groups relative to the Control group. This enrichment aligns with a study by Segal et al. (56), which demonstrated that the enrichment of oral bacteria, such as *Prevotella* and *Veillonella,* in the lungs can accelerate lung inflammation. Specifically, at the species level, *Prevotella melaninogenica* was enriched in both the TB and TBCOVID groups. Furthermore, the TB group also showed enrichment of other species of *Prevotella,* namely *Prevotella pallens*.

*Alloscardovia omnicolens* enriched in the TB and COVID groups has been identified as a causative agent of empyema in a patient with suspected TB (57). Interestingly,

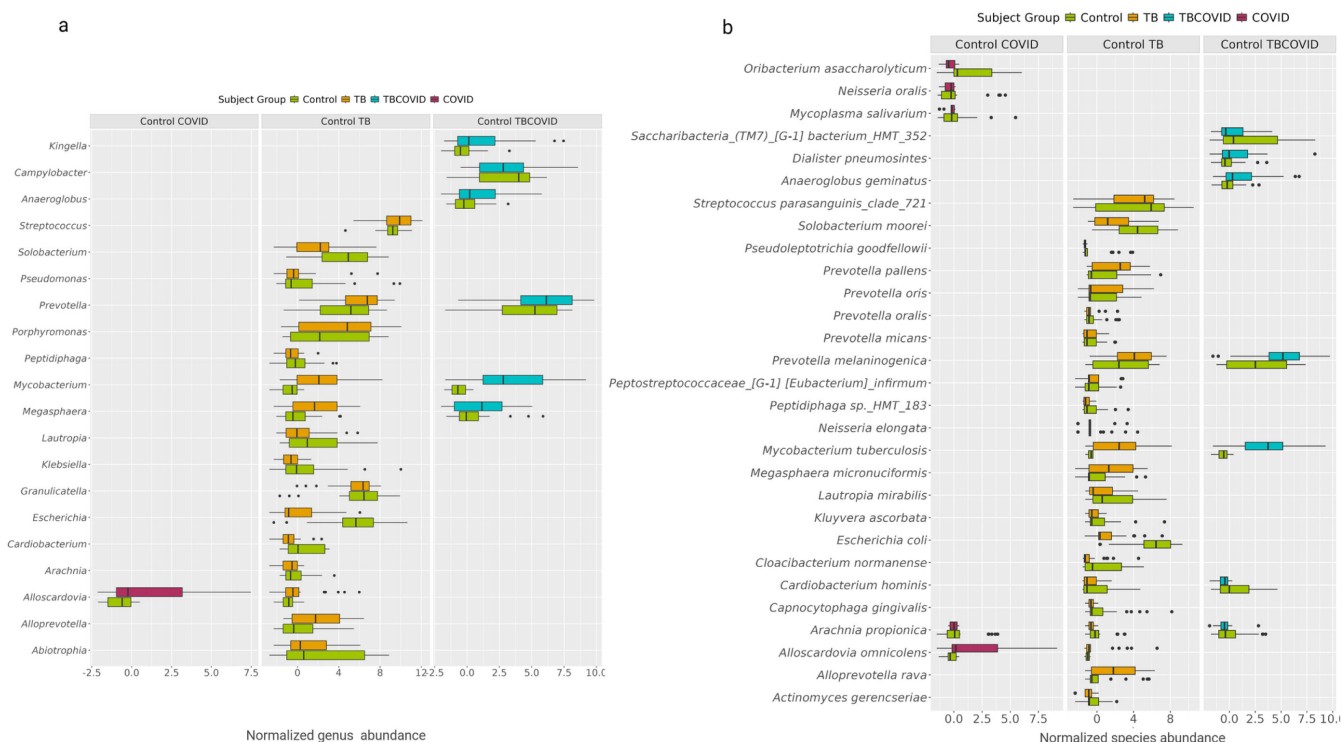

**FIG 4** Disease vs Control group comparisons. The plots show the distribution of normalized abundances of genera (Plot a) and species (Plot b) that are up- or down-regulated in disease relative to the Control group. Only genera/species selected by at least two of the methods (ANCOM-BC, LinDA, and corncob) at FDR <0.2, or identified as structural zeros by ANCOM-BC, are shown (see Supplementary Files D4 and D5 for the *P* values, adjusted *P* values, and LFC for each genus/species).

species like *Oribacterium asaccharolyticum*, *Neisseria oralis,* and *Mycoplasma salivarium* were found to be depleted in the COVID group. This depletion could be attributed either to the limited number of samples in the COVID group (only 41%) compared to the other groups, or it could be related to the disease itself, as reported in other studies (58–60). Similarly, the species *Arachnia propionica* was not observed in any of the samples in the TB and COVID groups.

The TB group was also enriched with the opportunistic pathogen *Megasphaera micronuciformis,* which has been associated with respiratory tract infections (61, 62); the genus *Porphyromonas* enriched in TB also has a similar association (63). *Anaeroglobus geminatus*, a pathogen linked to periodontitis and chronic obstructive pulmonary disease, was enriched in the TBCOVID group (64). These DA genera/species that are consistent with findings from other related studies increase confidence in our generated metagenomic data.

## Metagenome-related effects of COVID-19 on TB

### Altered microbial composition in TB with COVID-19 co-infection

As our main focus was to understand microbiome alterations in the presence of COVID-19 in TB patients, we next performed differential analysis between the TB and TBCOVID groups, both at genus and species levels (Fig. 5). At the genus level, seven genera were depleted, and none were enriched in the TBCOVID group. At the species level, five species were depleted, and four were enriched.

We then investigated the pathogenic status or inflammation-inducing potential of DA species to understand their potential roles in disease. Among the species found to be depleted in the TBCOVID group was *Lancefieldella parvula*, a common resident of oral cavity that rarely causes infections (65). *Prevotella pallens*, a species of a genera with links to inflammation (56), also showed reduced abundance. On the contrary, *Rothia mucilaginosa,* which has demonstrated an inhibitory effect on pro-inflammatory cytokines (66) both *in vitro* and *in vivo,* was also depleted, suggesting that it may have a negative effect on lung inflammation. Other species that were slightly down-regulated in the TBCOVID group include *Stomatobaculum longum* and *Saccharibacteria (TM7) [G-3] bacterium HMT 351*.

Inspection of the pathogenicity or inflammatory potential of the four species detected as enriched in the TBCOVID group, viz., *Veillonella parvula*, *Prevotella melaninogenica*, *Escherichia coli*, and *Capnocytophaga gingivalis* yielded the following observations. As mentioned before, the genera *Veillonella* and *Prevotella*, although generally

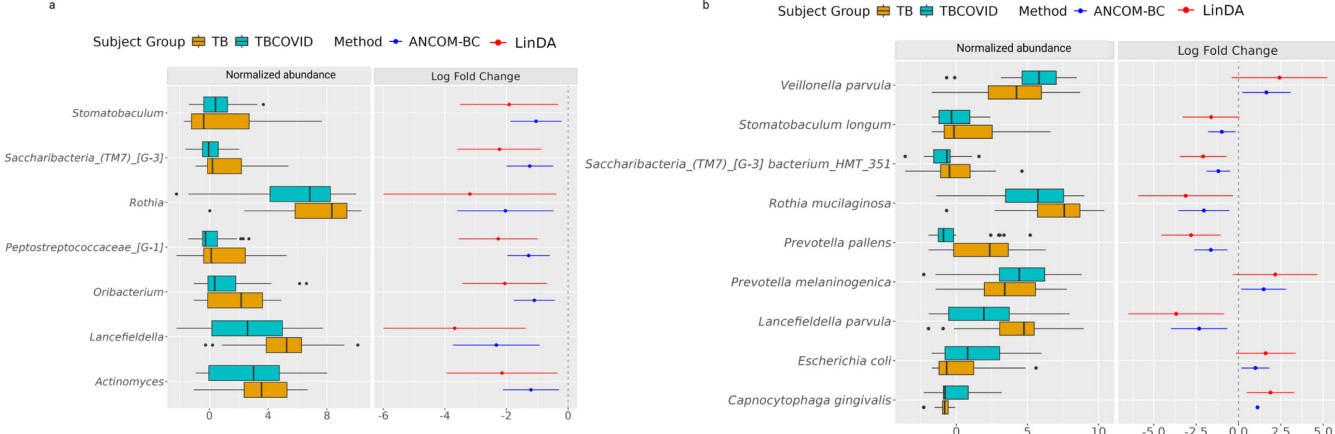

**FIG 5** Significant changes in the microbial composition of TBCOVID group when compared to TB group. The plots show the distribution of normalized abundances of genera (Plot a) and species (Plot b) that are up- or down-regulated in TBCOVID group relative to the TB group. Only taxa selected by at least two of the methods (ANCOM-BC, LinDA, and corncob) at FDR <0.2, or identified as structural zeros by ANCOM-BC, are shown (see Supplementary Files D4 and D5 for the *P* values, adjusted *P* values, and LFCs for each genus/species). As a corncob does not give LFC, only the LFCs from ANCOM-BC and LinDA are shown.

commensal, can induce cytokine production under dysbiotic conditions (56). *Escherichia coli*, which is commonly found in the intestines, has also been associated with the lung microbiome of lung cancer patients (67, 68). Finally, *Capnocytophaga gingivalis*, which was observed only in the TBCOVID group, has been associated with tumor growth in oral squamous cell carcinoma (69), and also with fulminant sepsis in a patient with COVID-19 (70). These results suggest that in TB patients, the co-infection of COVID-19 tends to increase the abundance of species with pathogenic or inflammation-inducing tendencies. Once replicated in further studies, these newly identified DA species/genera can lead to therapies addressing this dysbiosis associated with TBCOVID co-infection.

## Functional pathways and their implications for lung health

To gain deeper insights into the impact of COVID-19 on TB, we further examined enriched pathways between the TB and TBCOVID groups to determine whether the differentially abundant genera/species play a significant role at the functional level. The 30 pathways identified by at least two methods (ANCOM-BC, LinDA, or corncob) at FDR <0.01 and LFC ≥1.5 by LinDA were all up-regulated (see Fig. 6). For each pathway inferred by PICRUSt2, we also identified the taxa associated with it. With respect to the enriched pathways, the top contributing genera/species with at least 30 supporting reads and associated with a minimum of four pathways include *Escherichia coli*, *Eikenella corrodens*, *Lautropia mirabilis*, *Haemophilus* (*Haemophilus parainfluenzae*), *Neisseria*, *M. tb*, *Prevotella* (*Prevotella melaninogenica*), *Veillonella*, *Stenotrophomonas maltophilia*, *Actinomyces*, and *Selenomonas* (*Selenomonas sputigena*). The species *Stenotrophomonas maltophilia* and *Haemophilus* (*Haemophilus parainfluenzae*) belong to the Gammaproteobacteria family, which is known to thrive in inflammatory conditions (71). *Prevotella melaninogenica*, *Veillonella parvula,* and *Escherichia coli* were also significantly enriched in the TBCOVID group in the earlier analysis.

We then assessed the significance of the DA pathways in terms of their disease associations and functional roles. The presence of genera such as *Neisseria* and *Haemophilus* in the lungs may influence the Th1 response (72), and an imbalance in Th1/Th2 responses could lead to chronic lung inflammation (73). Both *Neisseria* and *Haemophilus,* along with other bacteria, such as *M. tb* and *Stenotrophomonas maltophilia,* were associated with cytidine diphosphate diacylglycerol (CDP-DAG) biosynthesis I and II, 6-hydroxymethyl-dihydropterin diphosphate biosynthesis III (Chlamydia), and phosphatidylglycerol synthesis-related pathways. The CDP-DAG pathways are involved in

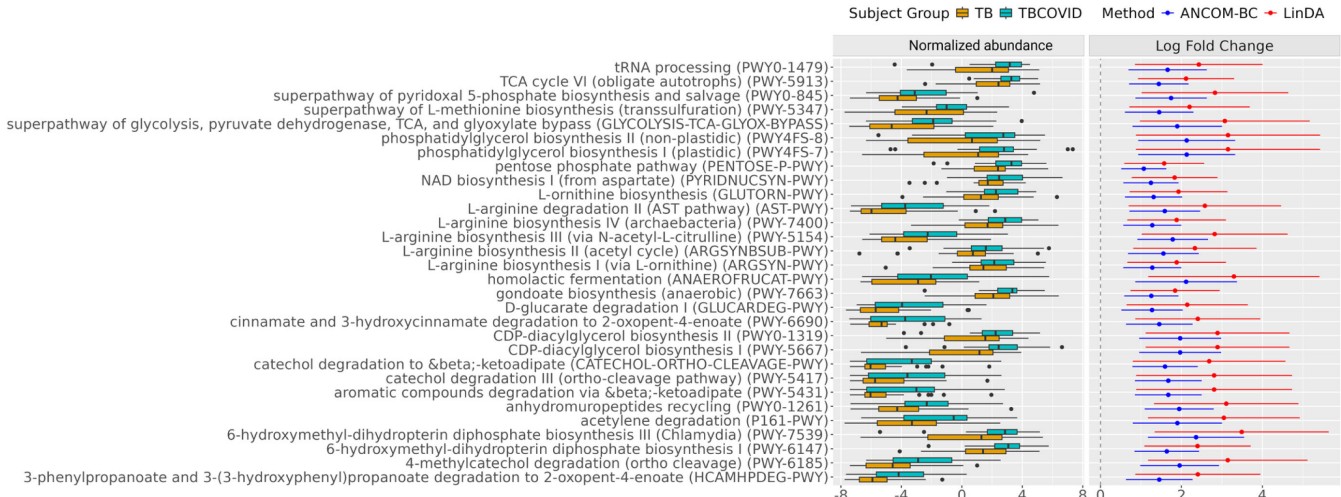

**FIG 6** Differential pathways between the TB and TBCOVID groups. The plot shows the distribution of normalized abundances of pathways that are up-regulated in the TBCOVID group relative to the TB group. Only pathways selected by at least two of the methods (ANCOM-BC, LinDA, and corncob) at FDR <0.01, or identified as structural zeros by ANCOM-BC, are shown (see Supplementary File D6 for the *P* values, adjusted *P* values, and LFCs for each pathway). As a corncob does not give LFC, only the LFCs from ANCOM-BC and LinDA are shown.

the generation of phospholipids and play a significant role in bacterial cellular processes (74, 75). Interestingly, the pentose phosphate pathway up-regulated in TBCOVID was also reported to be more abundant in COVID-19 patients compared to non-COVID controls (76).

Amino acid synthesis-related pathways such as superpathway of L-methionine biosynthesis, many L-arginine-related pathways, L-ornithine biosynthesis, and NAD biosynthesis I (from aspartate) were significantly enriched in the TBCOVID group. These pathways play a vital role in facilitating the growth (77, 78) and survival of bacteria under extreme conditions. Notably, *Stenotrophomonas maltophilia*, an opportunistic, multidrug-resistant pathogen associated with a wide range of infections, was found to be linked to 7 out of the 30 highly significant pathways. Other carbon metabolic pathways, such as tricarboxylic acid cycle, superpathway of pyridoxal 5′-phosphate biosynthesis and salvage, and other degradation-related pathways, were also up-regulated in the TBCOVID group (see Fig. 6). We have thus provided the disease and functional context of several pathways altered under COVID-19 co-infection in TB patients. The hypothesized mechanisms that the differentially abundant species may exhibit, along with their corresponding clinical outcomes arising from the upregulation of related pathways, are summarized in Table S3.

Besides DA pathway analysis, we also performed differential association analysis using the species abundances data—this analysis did not reveal any significantly altered associations between the TB and TBCOVID groups; a few associations, such as *Rothia mucilaginosa- Granulicatella adiacens* and *Dialister invisus-Dialister pneumosimtes,* remain in both groups (see Supplementary section Network and interaction analysis and Fig. S6 for details). Overall, the results of the differential genera/species and functional pathway analysis suggest that COVID-19 co-infection in TB may accelerate dysbiosis, promoting the colonization of pathogenic/inflammatory bacteria.

## Analysis of *M. tb* genomes from TB and TBCOVID groups

Moving beyond sputum microbiome analysis, we wanted to inspect mutations within *M. tb* isolates from the TB and TBCOVID groups and assess differences, if any, in these mutations from the two groups. We analyzed WGS data of 9 TB and 11 TBCOVID samples, and the variants/mutations called from these genomes using TB-Profiler and Demixer (see Materials and Methods). Based on the catalog of drug-resistant mutations in TB-Profiler, all samples contained at least one drug-resistant mutation, with the exception of two in the TBCOVID group. Through analysis of all detected mutations and reads supporting these mutations, Demixer assigned all WGS TB samples to lineage 1, except one which was assigned to lineage 4. The TBCOVID samples were also predominantly assigned to lineage 1 (eight samples), followed by lineage 2 (two samples) and lineage 4 (one sample). The phylogenetic tree also did not reveal any specific grouping of TB/TBCOVID samples (see Fig. S7), indicating that there are no clear genetic differences between the two groups. This is to be expected given that *M. tb* is a slowly evolving bacterium compared to SARS-CoV-2, which is a rapidly evolving virus with shorter onset and recovery periods (typically less than 20 days [79]), limiting the opportunity for introducing genetic alterations in the TB genome during co-infection.

## Integrating microbiome information with clinical adverse outcomes for personalized treatments

Our focus so far has been on group-level analyses (e.g., comparing taxa/pathway abundances between the TB vs TBCOVID groups) rather than individual sample-level analyses. To facilitate personalized tailoring of antibiotics/treatment strategies, we performed a sample-level clustering analysis of all TB and TBCOVID samples in order to identify subsets of individuals with similar microbial pathway activities and subsequently examine their follow-up clinical status (treatment, drug resistance, and single/multiple episode information obtained from the Nikshay portal on 26 February 2025, which is several years after the year 2021 when sputum samples were collected). Importantly,

all the observations in this analysis are only associative and do not imply predictive relationships between microbial signatures and outcomes. Specifically, we hierarchically clustered samples using the CLR-transformed abundances of only the DA pathways (30 TBCOVID-elevated pathways reported above), as shown in Fig. 7a. We observed that 70% of the TB samples were assigned to cluster 1, whereas 75% of the TBCOVID samples belonged to cluster 2, as expected since the clustering is based on the DA pathways. Biosynthesis-related pathways showed greater enrichment (more pronounced with darker colors in Fig. 7a) in cluster 2 compared to cluster 1.

To examine the clusters further in terms of clinical conditions, we call a patient as having "adverse outcome" if the patient experienced at least one of the following conditions during the period from 2021 to 2025, as documented in the Nikshay portal:

- Multiple TB episodes: when a patient had more than one episode of TB during the follow-up period.
- Drug-resistant: when the patient was resistant to at least one TB drug (e.g., Rifampicin, Isoniazid).
- Treatment failure: when TB treatment did not clear the infection as expected.
- Death: mortality of the patient during the course of TB treatment.

Relevant details on the nature of adverse outcomes for all patients in the TB and TBCOVID groups are provided in Supplementary File D7, with corresponding statistics shown in the top right corner in Fig. 7a.

Seven individuals in cluster 2 had adverse outcomes, compared to three in cluster 1. Within cluster 2, we also identified a sub-cluster consisting of three individuals (SM026, SM038, and SM048), all of whom had experienced multiple TB episodes with stronger pathway enrichment. When looking for a pathogen in the respiratory pathogen catalog (see Materials and Methods) within the samples, we found *Stenotrophomonas maltophilia* in the sputum microbiomes of SM026 and SM048 (see Fig. 7b). *Stenotrophomonas maltophilia* has previously been linked to multidrug-resistant TB (81), and tailoring the antibiotic treatment to tackle both *M. tb* and *Stenotrophomonas maltophilia* may improve treatment outcomes for these patients. In general, for any patient, the presence of a concomitant pathogen along with TB (see Fig. 7b) can accelerate disease severity, and administering personalized drugs based on the detected pathogens in an individual can improve recovery, as reported in reference 82. This personalization can be more applicable to individuals in cluster 2 than cluster 1, as the number of pathogens from our pathogen catalog detected in cluster 2 is twice as high as in cluster 1 (see Fig. 7b).

## DISCUSSION

TB and COVID-19 are primarily lung infections, and understanding the impact of their co-infection on lung microbiome is essential for handling similar outbreaks in the future. Several studies have analyzed metagenomic data related to TB and COVID-19 separately (3, 4, 13–16). However, to date, no metagenomic study has investigated co-infection involving both diseases. Given that both are respiratory conditions, a metagenomic study is crucial to explore alterations in the respiratory microbiome under co-infection, potential microbial interactions, and their implications on human health. In this study, we have generated and comprehensively analyzed the metagenomic data set of sputum specimens from patients infected with both TB and COVID-19, as well as from groups with TB-only, COVID-only, and without both diseases. The key findings and limitations of our study are summarized below.

### Summary of findings in the context of related works

On comparing *M. tb* read counts between the TB and TBCOVID groups after quality checks, we found that the median of mycobacterial read count was higher in TBCOVID patients with 3+ positive severity (based on smear grade test). This highlights the effectiveness of utilizing metagenomic sequencing (83) for TB diagnosis. The TB and

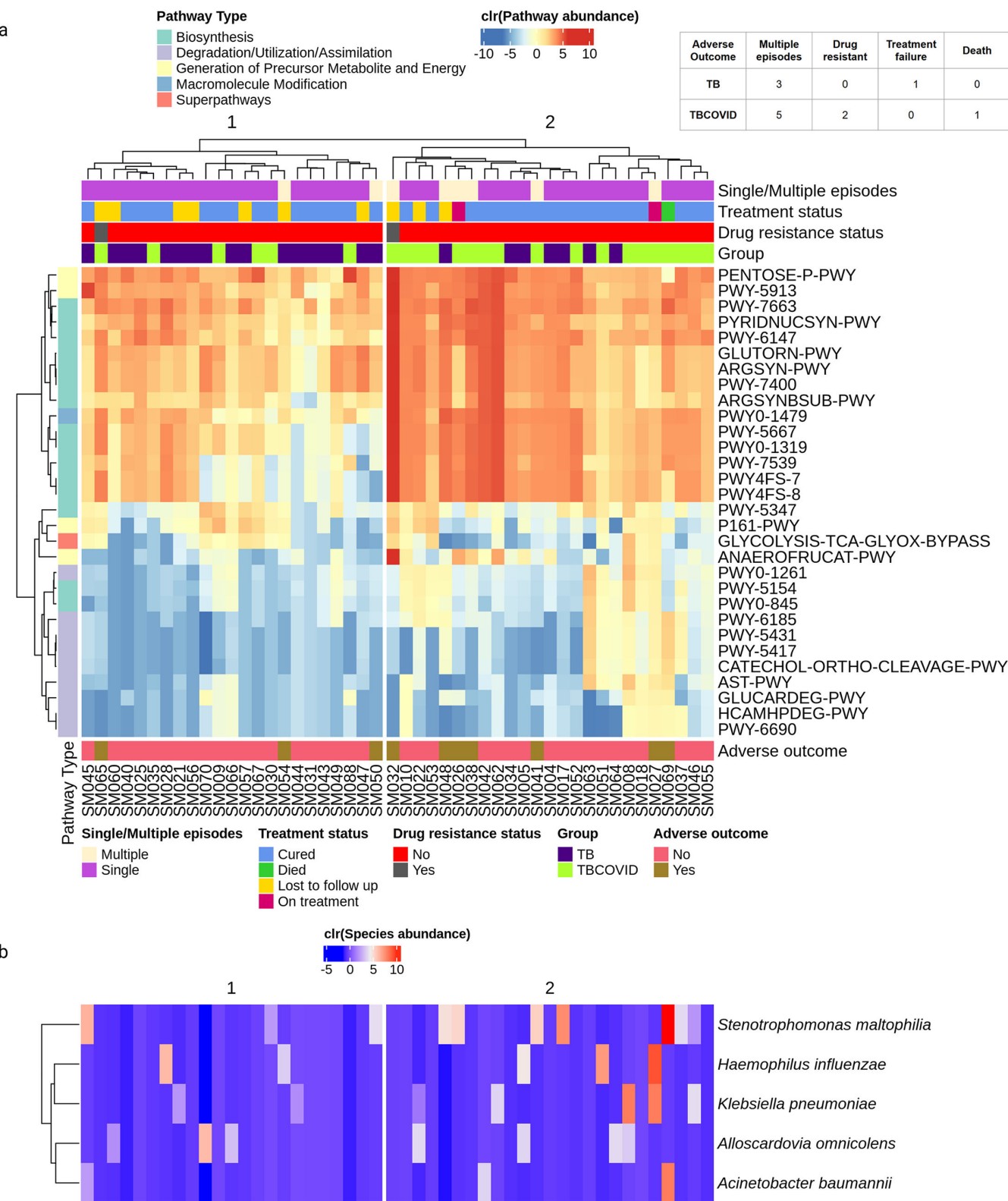

**FIG 7** Heatmap of microbial pathway activities overlaid with clinical/pathogen status of individuals. (a) Heatmap showing the PICRUSt2-reported abundance of DA pathways (which were elevated in TBCOVID relative to TB group), generated using the ComplexHeatmap package (80) in R. The samples were hierarchically clustered using complete linkage method. Cluster 1 is enriched with TB samples, while cluster 2 contains more TBCOVID samples. The color bars at the top indicate clinical status and disease grouping, while the color bar on the left indicates the top-level description of the pathways. (b) Heatmap showing the abundances of respiratory pathogens, with samples following the same order as in plot a.

TBCOVID samples were significantly enriched with *Prevotella*, *Megasphaera,* and *Mycobacterium* genera compared to the Control group, but the COVID-19 samples did not show any such enrichment, except for *Alloscardovia*. The microbial diversity of the COVID group was also lower than that of the other groups. It is important to note that this observation is subject to certain limitations, as described in the subsection below.

DA analysis between the TB and TBCOVID groups identified enrichment of species, such as *Prevotella melaninogenica*, *Veillonella parvula*, and *Capnocytophaga gingivalis,* in the TBCOVID group. These species have also been reported to be enriched in other studies involving COVID-19 patients (60). Though *Prevotella melaninogenica* and *Veillonella parvula* are commensals of the oral flora, their enrichment in the lower respiratory tract raises concern due to their association with pulmonary infections in humans (84). In particular, *Veillonella* has been identified as a highly adaptive organism capable of promoting the growth of other pathogens in a biofilm setting (85). The identification of these opportunistic pathogens also elucidates the potential of metagenomic sequencing in assessing the severity of pulmonary infections, as discussed in reference 86. Though metagenomic sequencing offers valuable insights, it is equally crucial to experimentally validate the identified species and their interactions in order to confirm our findings and develop personalized therapies.

*Escherichia coli,* a common intestinal bacterium belonging to the *Enterobacteriaceae* family, has recently been implicated in community-acquired pneumonia infection (87), and its levels were found to be elevated in the TBCOVID group than the TB group. An outbreak of NDM-5-producing *Escherichia coli* in COVID-19 patients has been reported in reference 88. Interestingly, *Escherichia coli* levels were significantly lower in both the TB and TBCOVID groups when compared to the Control group. However, as the Control group lacked associated covariates, the health status of individuals in this group could not be verified (see subsection below for limitations). Another perspective on the prevalence of *Escherichia coli* in sputum can be linked to the concept of "gut-lung axis," which suggests that intestinal dysbiosis can lead to the translocation of *Escherichia coli* to lungs by compromising the intestinal barrier (89). These insights highlight the potential disruption of the gut microbiome and highlight the need for interventions that promote both lung and gut health.

Our pathway analysis sheds light on the up-regulation of different pathways of the bacterial community in the lung microbiome. Specifically, the CDP-DAG pathways resulting from pathogens, such as *Haemophilus influenzae* and *Stenotrophomonas maltophilia,* were up-regulated. Notably, the effective functioning of the lungs relies largely on phospholipids, and dysregulation of the related pathways has been implicated in many pulmonary diseases (90). While our study does not provide direct insights into the role of bacterial CDP-DAG pathways on lung phospholipids, future experiments exploring this connection would be valuable. WGS analysis did not reveal any genetic differences between the TB and TBCOVID groups. We also explored integrating clinical information with microbial pathway activities to identify sub-clusters of high-risk individuals exhibiting similar pathway activities and having respiratory pathogens other than *M. tb*. Targeted/personalized antibiotic interventions based on the concomitant pathogens detected in these individuals may improve treatment outcomes.

## Limitations and future work

One major limitation of our study is the moderate-to-small data set size, reducing the statistical power and generalizability of the diversity and differential abundance findings. Nevertheless, power calculations based on PERMANOVA (at species level) using the micropower package in R indicated sufficient power (>99%) for distinguishing groups comprising 24 samples each, but moderate power (57%) for groups with only 10 samples (91). Power analysis for species-level differential abundance, performed using the powmic package in R, indicated moderate statistical power ($\approx$60%), suggesting that the observed diversity and differential abundance patterns are fairly robust (92). Moreover, as all samples were collected from a single region, the cohort may not adequately

represent a diverse population, introducing potential selection bias. To further assess the robustness of our findings, we analyzed two independent cohorts from different geographical regions (see Supplementary section Assessment of generalizability across independent cohorts and Tables S4 to S7). Across the Control vs COVID, Control vs TB, and Control vs TBCOVID comparisons, we observed consistent patterns in a subset of differentially abundant taxa across cohorts, suggesting that these associations are robust and unlikely to be due to spurious chance.

The smaller number of COVID-19 samples compared to the other groups may also have contributed to the lower microbial diversity observed in the COVID group. Moreover, the lack of covariate information for the Control and COVID groups limits the understanding of potential confounding factors influencing microbial composition. In addition, the non-availability of metagenomic data after treatment prevents us from assessing how TB treatment affects the microbiome and the relative changes in microbial communities before and after treatment. Despite these limitations, our study provides insights into the taxa altered under both single and joint infections of TB and COVID-19. It is important to note that this is the first study investigating double diseases involving TB and COVID-19, and the findings, taken together, offer a set of hypotheses on the role of pathogens in TB progression and treatment outcomes. Future studies could focus on experimental validation and replication of these findings in large multi-center cohorts. In addition, conducting longitudinal studies could provide a better understanding of disease progression and treatment effects, and genetic analyses examining variations in *M. tb* over long time periods may help answer the question of whether co-infection influences bacterial evolution.

## Conclusion

In this study, we performed metagenomic analysis on sputum samples to study the impact of COVID-19 on the lung microbiome of patients with tuberculosis. Using different metagenomic, pathway, and WGS analyses, our study reveals new insights into altered microbial composition in patients with TBCOVID. Our findings highlight the scope of metagenomic sequencing in TB diagnosis to identify co-infecting pathogens. The identification of up-regulated phospholipid-related pathways also provides valuable information on the role of different pathogens in the presence of TB and holds potential for broader applications in precision medicine.

## ACKNOWLEDGMENTS

This project was supported by funding from the Robert Bosch Center for Data Science and Artificial Intelligence, IIT Madras, through the SB21221740CSRBEI008892 Project. B.V.P. acknowledges funding support from Women Leading IIT Madras (WLI) (SB24250033CSIITM008892).

We thank members of our BIRDS (Bioinformatics and Integrative Data Science) and Centre for Integrative Biology and Systems Medicine research groups for their valuable input.

## AUTHOR AFFILIATIONS

[1]Department of Computer Science and Engineering, Indian Institute of Technology (IIT) Madras, Chennai, India
[2]Center for Integrative Biology and Systems Medicine, IIT Madras, Chennai, India
[3]Robert Bosch Centre for Data Science and Artificial Intelligence, IIT Madras, Chennai, India
[4]ICMR National Institute for Research in Tuberculosis (NIRT), Chennai, India

## AUTHOR ORCIDs

Brintha Vijayakumar Padmavathy http://orcid.org/0009-0002-2300-8171
Sivakumar Shanmugam http://orcid.org/0000-0001-9203-2664

Manikandan Narayanan  http://orcid.org/0000-0002-8490-4087

## FUNDING

| Funder | Grant(s) | Author(s) |
|---|---|---|
| Women Leading IIT Madras (WLI) | SB24250033CSIITM008892 | Brintha Vijayakumar Padmavathy |
| Robert-Bosch Center for Data Science and Artificial Intelligence | SB21221740CSRBEI008892 | Manikandan Narayanan |

## AUTHOR CONTRIBUTIONS

Brintha Vijayakumar Padmavathy, Conceptualization, Data curation, Formal analysis, Funding acquisition, Investigation, Methodology, Software, Validation, Visualization, Writing – original draft, Writing – review and editing | Ashok Kumar Shanmugavel, Data curation | Sivakumar Shanmugam, Conceptualization, Data curation, Formal analysis, Investigation, Methodology, Project administration, Resources, Supervision, Validation, Visualization, Writing – review and editing | Manikandan Narayanan, Conceptualization, Data curation, Formal analysis, Funding acquisition, Investigation, Methodology, Project administration, Resources, Supervision, Validation, Visualization, Writing – review and editing

## DATA AVAILABILITY

The metagenomic and WGS data generated as part of this study are submitted to NCBI under the bio project ID: PRJNA1248699. The supplemental data files related to the manuscript are available at this link https://drive.google.com/drive/folders/1o7sikxfRSLLFTl3VZUGB8rAAxy4tlANZ?usp=drive_link.

## ETHICS APPROVAL

This study is approved by the Institutional Ethics Committees of Indian Institute of Technology (IIT) Madras (IEC/2021-01/MN/08) and National Institute for Research in Tuberculosis (NIRT), Chennai (289/NIRT-IEC/2022).

## ADDITIONAL FILES

The following material is available online.

### Supplemental Material

**Supplemental materials (Spectrum02220-25-s0001.pdf).** Supplemental text, figures, and tables.

### Open Peer Review

**PEER REVIEW HISTORY (review-history.pdf).** An accounting of the reviewer comments and feedback.

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
