## [Reviewer comments · Microbiology Spectrum]

Microbiology Spectrum

Dissecting the effect of single- and co- infection of TB and COVID-19 pathogens on the sputum microbiome

Brintha VP, Ashok Kumar Shanmugavel, Sivakumar Shanmugam, and Manikandan Narayanan

Corresponding Author(s): Manikandan Narayanan, Indian Institute of Technology Madras

Review Timeline:

Submission Date:	July 21, 2025
Editorial Decision:	September 5, 2025
Revision Received:	November 4, 2025
Accepted:	November 19, 2025

Editor: Bo-young Hong

Reviewer(s): Disclosure of reviewer identity is with reference to reviewer comments included in decision letter(s). The following individuals involved in review of your submission have agreed to reveal their identity: Robert Matovu (Reviewer #1); Neeta Pradhan (Reviewer #3); Bingdong Zhu (Reviewer #4)

Transaction Report:

DOI: <https://doi.org/10.1128/spectrum.02220-25>

Re: Spectrum02220-25 (Dissecting the effect of single- and co- infection of TB and COVID-19 pathogens on the sputum microbiome)

Dear Dr. Manikandan Narayanan:

Thank you for the privilege of reviewing your work. Below you will find my comments, instructions from the Spectrum editorial office, and the reviewer comments.

Please thoroughly address the reviewer's comments and make the 16S rRNA sequencing data publicly available to ensure reproducibility. Currently, WGS (n=20) raw data are available, but not 16S data (n=82). In addition, remove the inaccessible link to the raw data.

Revision Guidelines

Sincerely,
Bo-young Hong
Editor
Microbiology Spectrum

Reviewer #1 (Comments for the Author):

This research study displays an interesting timely method or approach by applying systematic metagenomics and Whole Genome Sequencing (WGS) analyses to find microbial signatures associated with TB, COVID 19 and other their co-infections.

The use of four distinct groups that is TB only, COVID 19 only, co-infected and uninfected controls is appropriate for comparative analysis.

The Author correctly notes that the quality controls plus batch effect were done which is essential for ensuring data reliability and reproducibility in high put sequencing research findings

Reviewer #2 (Comments for the Author):

1. The authors mention limitations such as the small sample size, especially in the COVID-only group, and lack of detailed covariate data for controls. This should be moved to a dedicated and prominent "Limitations" subsection within the Discussion for transparency.
2. Discuss implications of possible selection bias, the representativeness of the cohorts, and how these issues could have influenced diversity analysis and differential abundance findings. Explicitly state that some findings are preliminary and require validation in larger, multi-center datasets.
3. Add a summary table in the Results or Supplementary section detailing the clinical and demographic characteristics (age, gender, BMI, treatments, co-morbidities) for all four groups. This aids in assessing comparability and the potential for residual confounding. Clarify explicitly the matching process-especially how PSM was adjusted when some covariate information was missing and how gender misclassification in PSM may impact results.
4. The pathway analysis identifies several upregulated pathways (e.g., lipid metabolism, amino acid biosynthesis), but the biological and clinical importance of these should be more deeply discussed. Are these findings corroborated in other datasets? What are the pathogenic mechanisms hypothesized? A summary diagram or table showing links between differentially abundant taxa, pathways, and clinical outcomes would make the functional results clearer.
5. More explicitly define "adverse outcomes" and provide supporting statistics (number/percent with treatment failure, relapse, resistance, death, etc.) across clusters in the heatmap/clinical outcome analysis. Discuss time to event, if available, or provide at least median follow-up time, which is important for outcome interpretation. The manuscript should state clearly whether the microbiome signatures at baseline predict outcomes, or whether the analysis is strictly associative/retrospective.

Reviewer #3 (Comments for the Author):

The study overall emphasizes the importance of co-microbial profiling in managing TB and COVID-19 co-infections and highlights the potential of metagenomic sequencing for precision medicine.

There are a few suggestions-

- o Increase in Sample Size could ensure robust findings.
- o Longitudinal study data would better understand disease progression and treatment effects.
- o Expand Genetic Analysis to investigate genetic differences in Mycobacterium tuberculosis over longer periods to assess potential impacts of co-infection on bacterial evolution
- o Include diverse patient cohorts (e.g., different geographic regions, age groups, and socioeconomic backgrounds) to ensure generalizability of findings
- o Investigate Gut-Lung Axis to study the gut microbiome alongside the lung microbiome to explore potential interactions and their role in disease severity
- o Include diverse patient cohorts (e.g., different geographic regions, age groups, and socioeconomic backgrounds) to ensure generalizability of findings
- o Link microbiome findings with detailed clinical outcomes
- o Analyze the microbiome after TB treatment to understand the influence of therapy

Reviewer #4 (Comments for the Author):

The manuscript reported an interesting metagenomic alterations of sputum specimens in TB, COVID-19 and TBCOVID groups. My main questions are as follow:

1. Are the samples enough to get the conclusion? How to evaluate it?
2. What is the concrete method to evaluate the smear grade based TB disease severity. Are they all smear acid fast positive?
3. Figure 3a labels are ambiguous; replacing 'type' and 'control' with defined group names and ensuring legend consistency will avoid confusion.
4. In Fig 7, a short description for the cluster 1 and 2 is needed.

Review of Dissecting the effect of single- and co- infection of TB and COVID-19 pathogens on the sputum microbiome.

The study overall emphasizes the importance of co-microbial profiling in managing TB and COVID-19 co-infections and highlights the potential of metagenomic sequencing for precision medicine.

Main findings of the study

The study's main findings are as follows:

1. Impact of COVID-19 on TB Microbiome.
2. Functional Pathway Alterations which are linked to lung health and bacterial survival under extreme conditions.
3. The TBCOVID group exhibited distinct microbial compositions compared to TB-only and Control groups.
4. Clinical Implications suggesting that personalized antibiotic regimens targeting both TB and co-infecting pathogens may improve treatment outcomes
5. The study highlights the importance of co-microbial profiling to personalize TB treatment, especially in cases of co-infection with COVID-19.

Drawbacks-

The drawbacks like small sample size, Lack of Covariates for Control Group, Limited Experimental Validation, Limited Experimental Validation and limited scope of genetic analysis have already been acknowledged by the authors.

Suggestions-

There are a few suggestions-

- 1 Increase in Sample Size could ensure robust findings.
- 2 Longitudinal study data would better understand disease progression and treatment effects.
- 3 Expand Genetic Analysis to investigate genetic differences in Mycobacterium tuberculosis over longer periods to assess potential impacts of co-infection on bacterial evolution
- 4 Include diverse patient cohorts (e.g., different geographic regions, age groups, and socioeconomic backgrounds) to ensure generalizability of findings
- 5 Investigate Gut-Lung Axis to study the gut microbiome alongside the lung microbiome to explore potential interactions and their role in disease severity
- 6 Include diverse patient cohorts (e.g., different geographic regions, age groups, and socioeconomic backgrounds) to ensure generalizability of findings
- 7 Link microbiome findings with detailed clinical outcomes
- 8 Analyze the microbiome after TB treatment to understand the influence of therapy.

Dear Editor Bo-young Hong,

We thank you and the reviewers for the careful evaluation and constructive comments on different aspects of our work. We have diligently addressed all the comments through new analyses and revisions of the manuscript, and a summary of these key analyses/revisions is provided below.

- A summary table linking differentially abundant taxa, pathways, and clinical outcomes has been added to highlight the functional significance of our findings.
- Additional power analyses pertaining to the beta diversity and differential abundance tests have shown that the sample sizes are reasonably adequate for drawing robust conclusions as detailed below.
- We conducted a new analysis using two external single-disease cohorts to assess the generalizability of our findings. The results indicate that a subset of the differentially abundant taxa identified in our study are consistent across these independent cohorts.
- We have revised the text to clarify adverse outcomes in the Results section, and a new subsection **Limitations and Future Work** has been added to the Discussion to clarify the study's limitations.

Our point-by-point responses to all reviewer comments are provided below in this **Response to Reviewers** document (in blue font), where we outline all the additional analyses and text revisions undertaken. The corresponding new analyses/results and textual clarifications are shown in yellow color in the Marked-Up manuscript. The page numbers indicated in the responses below refer to page numbers in the Marked-Up manuscript.

We believe that all reviewer comments have been fully addressed, and are pleased that the revisions have significantly improved the manuscript. We are open to address any further questions or clarifications and look forward to your response.

Thank you,
Manikandan Narayanan
(on behalf of all co-authors)

Point-by-point response:

> Editor

> Comments to the Author

1. Please thoroughly address the reviewer's comments and make the 16S rRNA sequencing data publicly available to ensure reproducibility. Currently, WGS (n=20) raw data are available, but not 16S data (n=82). In addition, remove the inaccessible link to the raw data.

Response:

We have carefully addressed all reviewers' comments below, and have ensured that all the relevant data are publicly available to ensure reproducibility. Specifically, the metagenomic (16S rRNA sequencing, n=82 and additional 10 repeat samples) and whole-genome sequencing (WGS, n=20) data generated in this study are now publicly available in NCBI (BioProject ID: PRJNA1248699). The previously inaccessible link has also been removed from the manuscript.

> Reviewer #1

> Comments to the Author

1. This research study displays an interesting timely method or approach by applying systematic metagenomics and Whole Genome Sequencing (WGS) analyses to find microbial signatures associated with TB, COVID 19 and other their co-infections.
2. The use of four distinct groups that is TB only, COVID 19 only, co-infected and uninfected controls is appropriate for comparative analysis.
3. The Author correctly notes that the quality controls plus batch effect were done which is essential for ensuring data reliability and reproducibility in high put sequencing research findings

Response:

We thank the reviewer for their valuable time in reviewing our manuscript and for these positive comments.

> **Reviewer #2**

> Comments to the Author

1. The authors mention limitations such as the small sample size, especially in the COVID-only group, and lack of detailed covariate data for controls. This should be moved to a dedicated and prominent "Limitations" subsection within the Discussion for transparency.

Response:

We thank the reviewer for pointing this out. As suggested, all the limitations have been moved to a dedicated subsection **Limitations and Future work** within the Discussion section (Page number 28).

2. Discuss implications of possible selection bias, the representativeness of the cohorts, and how these issues could have influenced diversity analysis and differential abundance findings. Explicitly state that some findings are preliminary and require validation in larger, multi-center datasets.

Response:

Your comments are well-taken. We had mentioned the need for such validation in independent cohorts in our original manuscript, and we have now expanded on the points raised by the reviewer about potential selection bias, cohort representativeness and sample size in the **Limitations and Future Work** subsection of the revised manuscript (Page number 28).

With regard to sample size, we would like to note that we have done power analyses of diversity and differential abundance tests, and these analyses show that our sample size is reasonably sufficient to discover the underlying disease signatures. We would also like to note that due to the timing of COVID-19 waves in Chennai and its subsequent decline, we were unable to collect additional samples for the COVID and TBCOVID groups, resulting in a smaller cohort. Nevertheless, we believe that our findings provide valuable insights into the microbial composition associated with TB–COVID co-infection.

3. Add a summary table in the Results or Supplementary section detailing the clinical and demographic characteristics (age, gender, BMI, treatments, co-morbidities) for all four groups. This aids in assessing comparability and the potential for residual confounding. Clarify explicitly the matching process—especially how PSM was adjusted when some covariate information was missing and how gender misclassification in PSM may impact results.

Response:

Thank you for this suggestion. We agree with the reviewer on the importance of providing covariates for all four groups, but we had access to covariate data only through the government-run TB portal called Nikshay (as mentioned in our **Methods** section – Page number 6). So we could extract and provide the covariate data for only the TB and TBCOVID groups (see Supplementary File D1a for these demographic and clinical characteristics, and Supplementary Table S1 for a cohort-level summary), and not the non-TB groups (i.e., COVID and Control). This limitation was previously noted in the Discussion section and has now been moved to the separate subsection **Limitations and Future Work**. Please note that this limitation doesn't affect our findings from our comparison of the TB vs. TBCOVID groups.

Furthermore, as covariate information was available only for the TB and TBCOVID groups, we used propensity score matching (PSM) for sample selection based on covariate distributions in these groups. The COVID group comprised only 10 samples, all of which were included, while a random selection was applied for the Control group. Details of the selection procedure were provided in the **Methods** section (Page number 7 and 8).

We had already clarified in Supplementary Figure S1 that gender misclassification is unlikely to be a concern, as the differentially abundant species in the TB vs. TBCOVID groups do not appear biased by the gender covariate. To provide further support, we have now included additional information showing the gender distribution in both the full and subsetting datasets. It is evident that post-PSM-selection, the distribution of males vs. females in the TB group is similar to that in the TBCOVID group. For convenience and quick reference, the corresponding figure and its description have been copied below from the manuscript.

a

	Before PSM (314 samples)			After PSM (48 samples)		
	All	TB	TBCOVID	All	TB	TBCOVID
Male	236 (75.4%)	213 (74.7%)	23 (82.1%)	41 (85.4%)	21 (87.5%)	20 (83.3%)
Female	77 (24.6%)	72 (25.3%)	5 (17.9%)	7 (14.6%)	3 (12.5%)	4 (16.7%)

Supplementary Figure S1 (with panel a added): Significance of Gender covariate: a) The distribution of males and females in the 314 samples used for matching, and in the 48 samples selected through PSM, is shown. In the full dataset before PSM, the ratio of males to females in the TB group differed from that in the TBCOVID group. However, after sample selection through PSM, the two groups showed similar males-to-females ratios. b) The heatmap shows species found to be differentially abundant between the TB and TB-COVID groups (see Fig. 5 in the main text for details). Gender is indicated in the top annotation bar.

4. The pathway analysis identifies several upregulated pathways (e.g., lipid metabolism, amino acid biosynthesis), but the biological and clinical importance of these should be more deeply discussed. Are these findings corroborated in other datasets? What are the pathogenic mechanisms hypothesized? A summary diagram or table showing links between differentially abundant taxa, pathways, and clinical outcomes would make the functional results clearer.

Response:

We agree with the reviewer’s suggestion to link the differentially abundant taxa to the upregulated pathways. We therefore added a new Supplementary Table S3 summarizing these taxa, their associated pathways, hypothesized mechanisms, and clinical outcomes, thereby highlighting their biological and clinical significance. This table is based on an extensive literature review and also includes supporting studies that have reported these taxa (shown below for convenience).

To address the other question on corroborating the findings in other datasets, we performed a new analysis using two independent cohorts from different geographical regions (a COVID cohort from China, and a TB cohort from Karnataka, India) to evaluate the generalizability of our Control vs. single disease findings. Since ours is the first study to generate a metagenomic dataset for the double-disease condition involving COVID-19 and TB (i.e., since no such combined-disease cohorts were available in other published studies), we focused on single-disease cohorts for the replication analysis. The results indicate that a subset of the differentially abundant taxa observed are consistent across cohorts (discussed in Supplementary Section **Assessment of generalizability across independent cohorts**).

Supplementary Table S3: Enriched differentially abundant taxa and their possible mechanisms in TB-COVID co-infection: The summary is populated based on extensive literature review.

Differentially enriched species	Associated Pathways	Hypothesized mechanisms	Clinical outcomes	Reported in COVID-only studies	Reported in TB-only studies
*Prevotella melaninogenica	Pentose phosphate, gondoate biosynthesis,	 Overexpression of Prevotella proteins may promote NF-κB 	 Increased inflammation leading to TB progression or 	[7, 9, 11]	[1, 2]

	6-hydroxymethyl-dihydropterin diphosphate biosynthesis, NAD biosynthesis pathways	 pathway activation [3] Increased generation of metabolites [4] Increased Th17 inflammatory response [6] 	 COVID-19 severity [3, 8] can lead to acute lung injury and pneumonia [5] 		
* Veillonella parvula	L-arginine synthesis related pathways	 Can play a role in activation of NF-κB signaling pathway 	 Can promote inflammation [10] 	[9, 11]	[2]
** Escherichia coli	phospholipid and membrane synthesis, amino acid synthesis and carbon metabolic pathways	 Generation of ammonia due to catabolism of aminoacids [14] Alteration of membrane lipid composition [15] 	 Can promote increased virulence and antimicrobial resistance [14] 	[12, 13]	
+ Capnocytophaga gingivalis	gondoate biosynthesis pathway			[16]	

*These taxa are part of the normal lung microbiota [17, 18], but disease conditions can disrupt their symbiotic relationship with the host and other microbes.

**A commensal member of the gut microbiota [19], but its presence in lungs can result in dysbiosis.

+Fewer than 10 reads support the associated pathways.

5. More explicitly define "adverse outcomes" and provide supporting statistics (number/percent with treatment failure, relapse, resistance, death, etc.) across clusters in the heatmap/clinical outcome analysis. Discuss time to event, if available, or provide at least median follow-up time, which is important for outcome interpretation. The manuscript should state clearly whether the microbiome signatures at baseline predict outcomes, or whether the analysis is strictly associative/retrospective.

Response:

We have duly noted your suggestion. We have now explicitly defined "Adverse outcomes" in Page number 25 and provided supporting statistics in the top right corner of the heatmap in Figure 7. In addition, we have added a new Supplementary File D7 containing relevant information on adverse outcomes in each patient.

Regarding the time to event, the median follow-up time for monitoring TB patients was 6 months, conducted independently by healthcare providers and recorded in the Nikshay portal. We have now elaborated on this in the **Sample collection, covariates' information, and**

selection procedure subsection within the Materials and Methods section (Page number 4 and 5).

We have also clarified the associative (i.e., non-predictive) nature of the microbiome signatures in Page number 23.

> Reviewer #3

We have duly noted the summary provided by Reviewer #3 in the attached PDF, as well as the specific comments addressed to the authors. We thank the reviewer for summarizing the findings of our study and for the valuable suggestions, which we have addressed point by point below.

> Comments to the Author

The study overall emphasizes the importance of co-microbial profiling in managing TB and COVID-19 co-infections and highlights the potential of metagenomic sequencing for precision medicine.

There are a few suggestions-

1. Increase in Sample Size could ensure robust findings.

Response:

We appreciate this valuable suggestion. While we acknowledge that increasing the sample size would improve the statistical power, our power analysis for diversity and differential abundance analyses indicates that the current sample sizes are reasonably sufficient to draw robust conclusions. Although we would have liked to include a larger number of samples, this was not possible due to practical constraints, such as the timing of the COVID-19 waves and the decline in COVID-19 cases during the study period. Nevertheless, the findings from this study provide a foundation for conducting future larger, multicentric studies. The limitations related to small sample size, along with details of the power analyses, have been added in the new subsection **Limitations and Future work** within the Discussion section of manuscript (Page number 28).

2. Longitudinal study data would better understand disease progression and treatment effects.

Response:

We agree with the reviewer on the importance of studying progression and treatment effects using longitudinal data. As this is the first study focusing on the double-disease condition of TB and COVID-19, we focussed on single-timepoint analysis to address the relevant research questions. However, we will take the reviewer's input into consideration when designing our

future studies, and have indicated this as future work in the **Limitations and Future Work** subsection (Page number 28).

3. Expand Genetic Analysis to investigate genetic differences in *Mycobacterium tuberculosis* over longer periods to assess potential impacts of co-infection on bacterial evolution

Response:

You have raised an interesting point on the evolutionary dynamics of *M. tb* during co-infection. Notably, one of the corresponding authors, Dr. Sivakumar Shanmugam, along with his colleagues at NIRT, has previously investigated the genetic characteristics of *M. tb* in the context of HIV co-infection and observed relapses involving the same bacterial strain (references added below). Similar studies involving other co-infections, such as COVID-19, with larger sample sizes can be carried out in follow-up investigations. These points have been mentioned as future work in the **Limitations and Future Work** subsection (Page number 28).

Ref 1: Narayanan S, Swaminathan S, Supply P, Shanmugam S, Narendran G, Hari L, Ramachandran R, Loch C, Jawahar MS, Narayanan PR. Impact of HIV infection on the recurrence of tuberculosis in South India. *The Journal of Infectious Diseases*. 2010 Mar 1;201(5):691-703.

Ref 2: Shanmugam S, Bachmann NL, Martinez E, Menon R, Narendran G, Narayanan S, Tripathy SP, Ranganathan UD, Sawleshwarkar S, Marais BJ, Sintchenko V. Whole genome sequencing based differentiation between re-infection and relapse in Indian patients with tuberculosis recurrence, with and without HIV co-infection. *International Journal of Infectious Diseases*. 2021 Dec 1;113:S43-7.

4. Include diverse patient cohorts (e.g., different geographic regions, age groups, and socioeconomic backgrounds) to ensure generalizability of findings

Response:

We understand the reviewer's concern on addressing the generalizability of findings using diverse cohorts. To ensure the robustness of our findings, we applied our pipeline to the following two independent cohorts:

- A COVID cohort comprising 44 healthy and 52 COVID samples from China [20].
- A TB cohort comprising 16 healthy and 50 untreated TB samples from Karnataka, India [21].

Please note that these external cohorts are single-disease cohorts as there were no double-disease cohorts involving TB and COVID-19 that we could find in the literature.

The results indicate that a subset of differentially abundant taxa identified in the comparisons between Control and disease groups exhibit consistent patterns across cohorts, supporting the

replication of our findings. Details of this additional analysis and the corresponding results are provided in the Supplementary Material (Section **Assessment of generalizability across independent cohorts**, and Supplementary Tables S4–S7)

5. Investigate Gut-Lung Axis to study the gut microbiome alongside the lung microbiome to explore potential interactions and their role in disease severity

Response:

As indicated in our response to the comment #2, this is one of the first studies focusing on the microbiomes of both TB and COVID-19; therefore, we primarily focused on characterizing the sputum microbiome. This is based on the rationale that changes in the lung microbiome can be inferred from sputum due to its connection with the oral cavity and respiratory tract. While we agree with the reviewer that sputum samples alone are insufficient to investigate the gut-lung axis, we would like to respectfully clarify that understanding the gut-lung axis is beyond the scope of this study. We can consider this aspect in our future studies.

6. Include diverse patient cohorts (e.g., different geographic regions, age groups, and socioeconomic backgrounds) to ensure generalizability of findings.

Response:

Please note that this has already been addressed in our response to the reviewer's comment #4 above.

7. Link microbiome findings with detailed clinical outcomes

Response:

Thank you for raising this important point. We have now added a new Supplementary Table S3 summarizing the differentially abundant taxa with expected clinical outcomes and also elaborated on adverse outcomes (Page number 25). We would also like to highlight that an analysis on **Integrating microbiome information with clinical adverse outcomes for personalized treatments** is already provided in the **Results** section.

8. Analyze the microbiome after TB treatment to understand the influence of therapy

Response:

As suggested, analyzing the microbiome post-TB treatment could indeed provide valuable insights – but this is beyond the scope of our study for reasons explained next, and we've added a point about this in the limitation section (Page number 28). As per the ethics-approved study design, we are permitted to collect a participant's sputum sample only during his/her first visit and not during follow-up visits (specifically, only clinical data can be collected during the follow-up visit; details on this has now been added to the **Sample collection, covariates' information, and selection procedure** subsection within the **Materials and Methods** section - Page number 4 and 5). Also as indicated earlier, since ours is the first metagenomic study involving both TB and COVID-19, we wanted to focus on single-timepoint analyses for this study.

> **Reviewer #4**

> Comments to the Author

The manuscript reported an interesting metagenomic alterations of sputum specimens in TB, COVID-19 and TBCOVID groups. My main questions are as follow:

1. Are the samples enough to get the conclusion? How to evaluate it?

Response:

Thank you for the thoughtful question. We acknowledge the importance of assessing whether the available sample size is sufficient to draw meaningful conclusions.

To address this, we conducted power analyses to assess the statistical power of PERMANOVA (related to beta diversity) and differential abundance tests. We found that, at the species level, PERMANOVA demonstrated high power (>99%), while differential abundance analyses showed moderate power (~60%) for groups comprising 24 samples each. In our dataset, all groups except the COVID group have a sample size of 24. This observation suggests that our sample sizes except for the COVID group are reasonably sufficient for identifying disease signatures. These results have been added in the dedicated subsection on **Limitations and Future Work** within the Discussion section (Page number 28).

Below is a detailed description of the power analyses conducted at two different levels:

i) **Group-level analysis:** Using the micropower R package [22], we estimated the statistical power for sample sizes ranging from 3 to 21 based on beta diversity measures. The analysis indicated that power was approximately 57% at a sample size of 10, but increased to 99% when the sample size reached 21 (see Figure 1). In our study, with the exception of the COVID group, all three other groups (TB, TBCOVID, and Control) had 24 samples each, suggesting that the

achieved sample sizes were adequate to attain sufficient statistical power, at least for a subset of the analyses conducted using PERMANOVA.

Figure 1: Power analysis using the micropower package: (a) Distribution of PERMANOVA p-values obtained across different sample sizes (1,000 runs each) at an FDR cutoff of 0.05. (b) Estimated power values for varying sample sizes.

ii) **Individual taxon-level analysis:** Using the powmic package in R [23], we assessed statistical power for differential abundance analysis based on Wilcoxon rank-sum tests. Powmic employs simulation, wherein 5% of taxa are randomly assigned a log-fold change of two in either the case or control group, to estimate the effect of sample size on identifying truly differentially abundant taxa at an FDR threshold of 0.05. The estimated true positive rate (TPR) for sample sizes ranging from 10 to 60 (based on 1,000 runs each at an FDR cutoff of 0.05) is shown in Figure 2. At a sample size of 24, the TPR was approximately 60%, indicating that our study had moderate power to detect truly differentially abundant taxa.

Figure 2: Power analysis using the powmic package: Estimated TPR across varying sample sizes on simulated datasets using Wilcoxon testing. TPR here can be interpreted as equivalent to statistical power.

2. What is the concrete method to evaluate the smear grade based TB disease severity. Are they all smear acid fast positive?

Response:

Thank you for this question. Acid-Fast Bacillus (AFB) sputum smear grading test was used to assess the TB load and the same has been updated in the manuscript (Page number 6). To answer your question on the concrete method to evaluate the smear grade based TB disease severity, we would like to point out that sputum smear grading test gives only a rough indicator of TB disease severity [24, 25], with 1+ smear grade being an approximate indicator for low severity, 2+ for moderate severity, and 3+ for high TB severity.

To address your other question, all the TB and TBCOVID participants selected for microbiome profiling in our study were AFB-positive and the corresponding smear grading related covariates are already provided in Supplementary File D1b.

3. Figure 3a labels are ambiguous; replacing 'type' and 'control' with defined group names and ensuring legend consistency will avoid confusion.

Response:

Thank you for the careful observation. We have now updated the figure to fix these ambiguities (Added below for your convenience).

Figure 3: Characterization of species diversity and composition in metagenomic sequencing data obtained from the four groups

4. In Fig 7, a short description for the cluster 1 and 2 is needed.

Response:

We have now updated the figure description to address this comment.

References:

- 1) Qin, Mingyang, et al. "Dysbiosis associated with enhanced microbial mobility across the respiratory tract in pulmonary tuberculosis patients." *BMC microbiology* 25.1 (2025): 499.
- 2) Cheung, Man Kit, et al. "Sputum microbiota in tuberculosis as revealed by 16S rRNA pyrosequencing." *PLoS one* 8.1 (2013): e54574.
- 3) Khan, Abdul Arif, and Zakir Khan. "COVID-2019-associated overexpressed *Prevotella* proteins mediated host-pathogen interactions and their role in coronavirus outbreak." *Bioinformatics* 36.13 (2020): 4065-4069.
- 4) Wong, Kendrew K., et al. "Microbial contribution to metabolic niche formation varies across the respiratory tract." *Cell Host & Microbe* (2025).
- 5) Charles, Angel, et al. "515 *Prevotella melaninogenica*-derived Metabolites Induce Immune Reprogramming of Human THP-1 Macrophages." *Journal of Burn Care & Research* 46.Supplement_1 (2025): S113-S113.
- 6) Segal, Leopoldo N., et al. "Enrichment of the lung microbiome with oral taxa is associated with lung inflammation of a Th17 phenotype." *Nature microbiology* 1.5 (2016): 1-11.
- 7) Lu, Sifen, et al. "Metatranscriptomic analysis revealed *Prevotella* as a potential biomarker of oropharyngeal microbiomes in SARS-CoV-2 infection." *Frontiers in cellular and infection microbiology* 13 (2023): 1161763.
- 8) Larsen, Jeppe Madura. "The immune response to *Prevotella* bacteria in chronic inflammatory disease." *Immunology* 151.4 (2017): 363-374.
- 9) Haran, John P., et al. "Inflammation-type dysbiosis of the oral microbiome associates with the duration of COVID-19 symptoms and long COVID." *Jci Insight* 6.20 (2021): e152346.
- 10) Gudowska-Sawczuk, Monika, and Barbara Mroczko. "The role of nuclear factor kappa B (NF- κ B) in development and treatment of COVID-19." *International journal of molecular sciences* 23.9 (2022): 5283.
- 11) Bourumeau, William, et al. "Bacterial Biomarkers of the Oropharyngeal and Oral Cavity during SARS-CoV-2 Infection." *Microorganisms* 11.11 (2023): 2703.
- 12) Cut, Talida Georgiana, et al. "A retrospective assessment of sputum samples and antimicrobial resistance in COVID-19 patients." *Pathogens* 12.4 (2023): 620.
- 13) Liu, Hans H., et al. "Bacterial and fungal growth in sputum cultures from 165 COVID-19 pneumonia patients requiring intubation: evidence for antimicrobial resistance development and analysis of risk factors." *Annals of clinical microbiology and antimicrobials* 20.1 (2021): 69.
- 14) Alteri, Christopher J., and Harry LT Mobley. "Escherichia coli physiology and metabolism dictates adaptation to diverse host microenvironments." *Current opinion in microbiology* 15.1 (2012): 3-9.
- 15) Rowlett, Veronica W., et al. "Impact of membrane phospholipid alterations in Escherichia coli on cellular function and bacterial stress adaptation." *Journal of bacteriology* 199.13 (2017): 10-1128.
- 16) Ma, Shengli, et al. "Metagenomic analysis reveals oropharyngeal microbiota alterations in patients with COVID-19." *Signal Transduction and Targeted Therapy* 6.1 (2021): 191.
- 17) Lamoureux, Claudie, et al. "*Prevotella melaninogenica*, a sentinel species of antibiotic resistance in cystic fibrosis respiratory niche?." *Microorganisms* 9.6 (2021): 1275.

- 18) Poppleton, Daniel I., et al. "Outer membrane proteome of *Veillonella parvula*: a diderm firmicute of the human microbiome." *Frontiers in microbiology* 8 (2017): 1215.
- 19) Tenaillon, Olivier, et al. "The population genetics of commensal *Escherichia coli*." *Nature reviews microbiology* 8.3 (2010): 207-217.
- 20) Wu, Yongjian, et al. "Altered oral and gut microbiota and its association with SARS-CoV-2 viral load in COVID-19 patients during hospitalization." *npj Biofilms and Microbiomes* 7.1 (2021): 61.
- 21) Hazra, Druti, et al. "The impact of anti-tuberculosis treatment on respiratory tract microbiome in pulmonary tuberculosis." *Microbes and Infection* 27.3 (2025): 105432.
- 22) Kelly, Brendan J., et al. "Power and sample-size estimation for microbiome studies using pairwise distances and PERMANOVA." *Bioinformatics* 31.15 (2015): 2461-2468.
- 23) Chen, Li. "powmic: an R package for power assessment in microbiome case-control studies." *Bioinformatics* 36.11 (2020): 3563-3565.
- 24) Tran, Xuan Thuy, et al. "High prevalence and risk factors of positive sputum smear in newly diagnosed pulmonary tuberculosis patients in Vietnam." *Le Infezioni in Medicina* 33.2 (2025): 212.
- 25) Kassa, Getahun Molla, et al. "Sputum smear grading and associated factors among bacteriologically confirmed pulmonary drug-resistant tuberculosis patients in Ethiopia." *BMC Infectious Diseases* 21.1 (2021): 238.

Re: Spectrum02220-25R1 (Dissecting the effect of single- and co- infection of TB and COVID-19 pathogens on the sputum microbiome)

Dear Dr. Manikandan Narayanan:

Your manuscript has been accepted, and I am forwarding it to the ASM production staff for publication. Your paper will first be checked to make sure all elements meet the technical requirements. ASM staff will contact you if anything needs to be revised before copyediting and production can begin. Otherwise, you will be notified when your proofs are ready to be viewed.

Sincerely,
Bo-young Hong
Editor
Microbiology Spectrum

Reviewer #1 (Comments for the Author):

All comments have been addressed

Reviewer #3 (Comments for the Author):

No further comments to the authors